# Sloppy morphological tuning in identified neurons of the crustacean stomatogastric ganglion

**Adriane G Otopalik\*[†], Marie L Goeritz[†‡], Alexander C Sutton[†§], Ted Brookings[¶], Cosmo Guerini, Eve Marder\***

Biology Department and Volen Center, Brandeis University, Waltham, United States

**Abstract** Neuronal physiology depends on a neuron's ion channel composition and unique morphology. Variable ion channel compositions can produce similar neuronal physiologies across animals. Less is known regarding the morphological precision required to produce reliable neuronal physiology. Theoretical studies suggest that moraphology is tightly tuned to minimize wiring and conduction delay of synaptic events. We utilize high-resolution confocal microscopy and custom computational tools to characterize the morphologies of four neuron types in the stomatogastric ganglion (STG) of the crab *Cancer borealis*. Macroscopic branching patterns and fine cable properties are variable within and across neuron types. We compare these neuronal structures to synthetic minimal spanning neurite trees constrained by a wiring cost equation and find that STG neurons do not adhere to prevailing hypotheses regarding wiring optimization principles. In this highly modulated and oscillating circuit, neuronal structures appear to be governed by a space-filling mechanism that outweighs the cost of inefficient wiring.

\*For correspondence: aotopali@
brandeis.edu (AGO); marder@
brandeis.edu (EM)

[†]These authors contributed
equally to this work

Present address: [‡]Department
of Marine Science, University of
Auckland, Auckland, New
Zealand; [§]Boston Consulting
Group, Boston, United States;
[¶]Q-State Biosciences,
Cambridge, United States

Competing interest: See
page 28

Reviewing editor: Ronald L
Calabrese, Emory University,
United States

## Introduction

Neuronal physiology arises from a neuron's ion channel distribution and its unique geometry. Numerous studies have shown that neurons and circuits can function despite remarkable variability in their ion channel composition (*Prinz et al., 2004*; *Schulz et al., 2006*; *Goaillard et al., 2009*; *Norris et al., 2011*; *Roffman et al., 2012*; *Gutierrez et al., 2013*; *Sakurai et al., 2014*; see *Marder and Goaillard (2006)*, *Calabrese et al. (2011)* and *Marder et al. (2015)* for reviews). These observations suggest that neurons can generate robust, highly conserved firing patterns despite some degree of sloppiness in their underlying physiological parameters.

In contrast, it is often assumed that precise neuronal computations and plasticity rules arise from specialized neuronal morphologies (*Mainen and Sejnowski, 1996*; *Stiefel and Sejnowski, 2007*; *Cuntz et al., 2010*). Experimental and theoretical studies have argued that optimal wiring principles determine neuronal structure during development (*Chklovskii, 2000*, *2004*; *Chen et al., 2006*; *Wen and Chklovskii, 2008*; *Cuntz et al., 2007*, *2010*, *2012*; *Kim et al., 2012*). Yet, surprisingly, few studies (*Goodman, 1976*, *1978*; *Miller and Jacobs, 1984*; *Wang et al., 2002*; *Cuntz et al., 2008*, *2010*) have reconstructed multiple neurons of the same type at high enough resolution to observe or quantify the stereotypy expected to arise during development, given such optimization principles. Thus, relatively little is known regarding the practical applications of such growth rules and the degree of morphological precision required to produce reliable neuronal physiology in varying circuit contexts.

Here, we present quantitative morphological analyses of macroscopic branching patterns and finer cable properties, within and across four identified neuron types in the stomatogastric ganglion (STG) of the crab, *Cancer borealis*, a small central-pattern generating circuit. The 26 neurons of the

crab STG have large somata (50–150 µm in diameter) situated around a dense neuropil region, wherein synaptic connections are made (*Maynard, 1971*; *Maynard and Dando, 1974*; *King, 1976a*, *1976b*; *Kilman and Marder, 1996*). STG neurons have complex, highly branched structures that ramify throughout the neuropil region (*Baldwin and Graubard, 1995*; *Wilensky et al., 2003*; *Bucher et al., 2007*; *Thuma et al., 2009*) and are subject to modulation by numerous peptides and amines released diffusely in the hemolymph and by descending modulatory inputs (*Christie et al., 1995a*, *1995b*, *1997*; *Goldberg et al., 1988*; *Marder et al., 1986*; see *Marder and Bucher (2007)* and *Nusbaum and Blitz, 2012* for reviews). The 14 identified STG neuron types are physiologically defined by their muscle innervation patterns (*Maynard and Dando, 1974*), highly conserved voltage waveforms, and participation in one or both of two rhythms: the fast, ongoing pyloric rhythm and the slow, episodic gastric mill rhythm (*Harris-Warrick et al., 1992*). STG circuit function is mediated predominately by slow oscillations (*Graubard and Ross, 1985*; *Ross and Graubard, 1989*) and graded inhibitory transmission (*Eisen and Marder, 1982*; *Marder and Eisen, 1984*; *Maynard and Walton, 1975*; *Graubard et al., 1980*; *Manor et al., 1997*, *1999*). In fact, phase relationships between STG neurons can be maintained in the absence of spikes (*Graubard, 1978*; *Raper, 1979*; *Graubard et al., 1983*; *Anderson and Barker, 1981*).

Recent work demonstrated that, despite their morphological complexity, STG neurons are electrotonically compact (*Otopalik et al., 2017*). Thus, the slow voltage events arising from graded synaptic transmission undergo minimal electrotonic decrement across the neuronal structure. Such electrotonic structures could feasibly mask animal-to-animal variability in neuronal morphology within identified STG neuron types. Additionally, given their compact electrotonic structures, it is plausible that developmental growth of STG neurons may not be constrained by wiring optimization rules, but perhaps other rules that speak to their role in this highly modulated, oscillating circuit. In the present study, we quantitatively characterize the morphology of four STG neuron types across several animals. Then, we investigate whether these identified neurons adhere to prevailing wiring optimization principles.

## Results

The neurons of the STG of the crab *Cancer borealis* each have a primary neurite that branches extensively throughout the neuropil, forming synapses. This primary neurite also gives rise to one or multiple axons that project to specific stomach muscles or anterior ganglia (*Maynard, 1971*; *Maynard and Dando, 1974*; *King, 1976a*, *1976b*; *Kilman and Marder, 1996*). *Figure 1A* shows an exemplar gastric mill (GM) neuron situated in the STG neuropil. The stomatogastric nerve (*stn*) contains dozens of descending neuromodulatory inputs that branch extensively throughout the neuropil and diffusely release a cocktail of peptides and amines (*Marder and Bucher, 2007*; *Nusbaum and Blitz, 2012*).The four neuron types examined here are motor neurons and send axons through either the anterior lateral nerve (*aln*) or dorsal ventricular nerve (*dvn*; *Figure 1B*). GM neurons (four copies in each STG) typically have 3–5 axons, with one or multiple axons projecting to the *aln*s and *dvn* (*Figure 1Bi*). In contrast, the lateral gastric (LG, one copy in each STG), pyloric dilator (PD; two copies in each STG), and lateral pyloric (LP, one copy in each STG) neurons project single axons down the *dvn* (*Figure 1Bii–iv*).

To quantify and compare macroscopic and finer morphological properties within and across these four neuron types (N = 4 of each type), we manually traced high-resolution confocal image stacks of their Lucifer Yellow dye-fills (see *Figure 1—figure supplement 1–4* for images of all 16 neurons) and then extracted the following measurements from either the confocal imaging stacks directly or skeletal reconstructions using a suite of quantitative morphology tools written in Python (see Materials and methods for in-depth description of our approach; all scripts are publicly available in the Marder Lab GitHub repository at github.org/marderlab). These data are presented in three sections: (1) Macroscopic Structure, which presents quantitative measures of somata size and location, hand-like features, and neuropil branching patterns; (2) Fine Structure, which describes neurite lengths, diameters, and branch point geometry—all features that are critical for the propagation of voltage signals across the neurite tree; and (3) an investigation of wiring optimization principles and whether they apply to STG neurons.

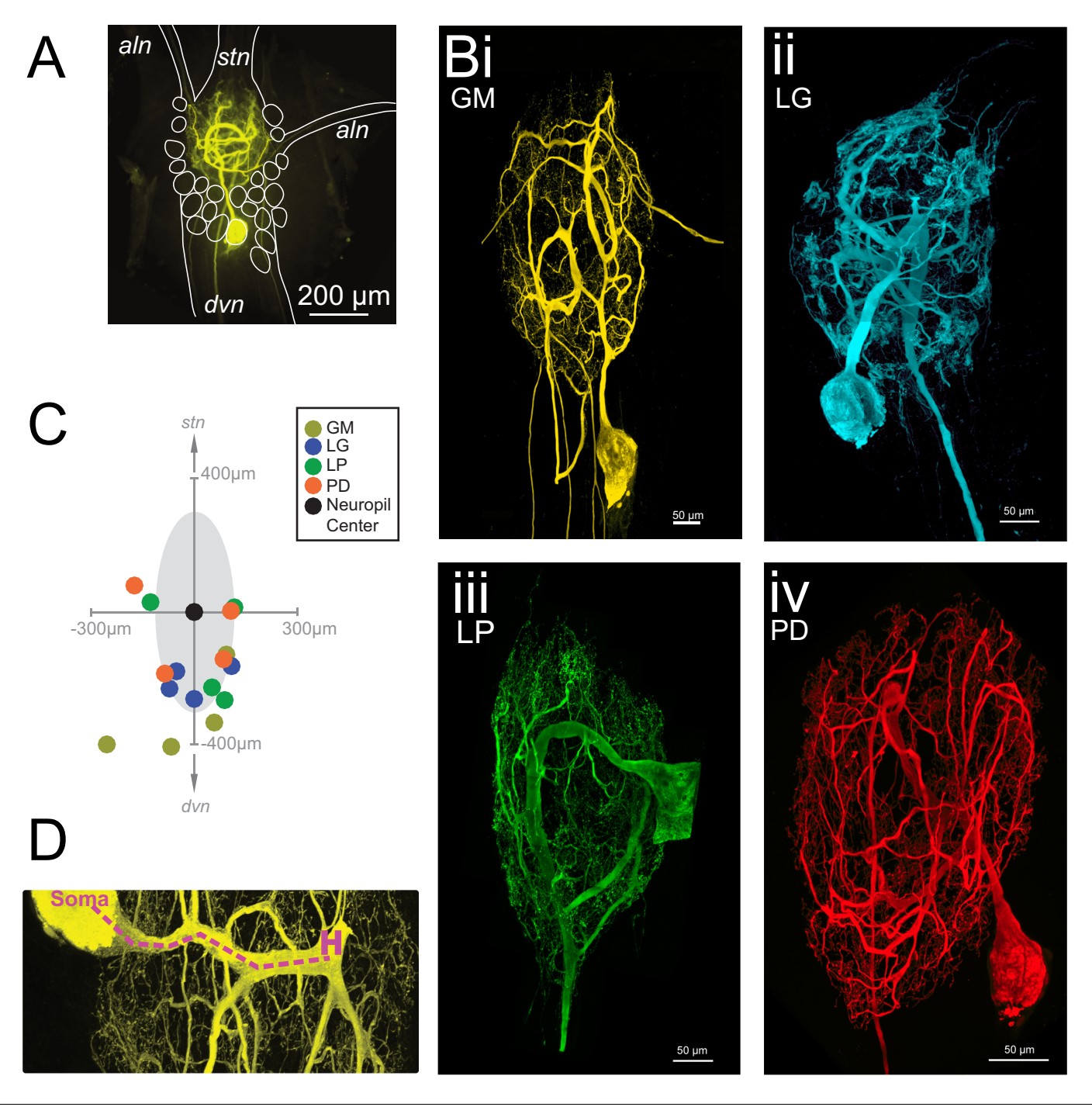

**Figure 1.** Macroscopic features of four identified neurons of the stomatogastric ganglion. Each neuron has a primary neurite that ramifies throughout the neuropil and sends one or more axons to peripheral muscles or ganglia via the anterior lateral (*aln*) or dorsal ventricular (*dvn*) nerves. (**A**) Schematic of a gastric mill (GM) neuronal dye-fill as situated in the intact ganglion and the *aln*, *dvn*, and stomatogastric nerve (*stn*). Other cell bodies are outlined in white. (**B**) Maximum-intensity projections of four Lucifer yellow dye-filled STG neurons. The GM neuron (**i**) has multiple axons that exit the neuropil bilaterally via the *aln*s, and posteriorly via the *dvn*. The lateral gastric (LG, **ii**), lateral pyloric (LP, **iii**), and pyloric dilator (PD, **iv**) neurons, respectively, each project only one axon that leaves the neuropil via the *dvn*. (**C**) Locations of identified somata across different preparations. Within each preparation, positions were normalized to the ganglion center-of-mass (at origin) and oriented with the stomatogastric nerve (*stn*) at the top and dorsal ventral nerve (*dvn*) at the bottom (as described in Materials and methods). (**D**) Image of neuronal dye-fill (yellow) with path (purple dashed line) from soma to the first multi-furcation, or 'hand-like' structure (**H**).

*Figure 1 continued on next page*

*Figure 1 continued*

The following figure supplements are available for figure 1:

**Figure supplement 1.** Fluorescent dye-fills of identified lateral pyloric (LP) neurons.

**Figure supplement 2.** Fluorescent dye-fills of identified pyloric dilator (PD) neurons.

**Figure supplement 3.** Fluorescent dye-fills of identified lateral gastric (LG) neurons.

**Figure supplement 4.** Fluorescent dye-fills of identified gastric mill (GM) neurons.

## I. Macroscopic structure

### Somata, hand- and hair-like structures

Each of the 16 neurons had a relatively large soma diameter (pooled mean ± SD = 125.8 ± 47.5 µm across all neurons). Earlier work in lobster, *Homarus americanus* and *Panulirus interruptus,* STG suggested some stereotypy in soma positions of certain neuron classes (*Bucher et al., 2007* and *Thuma et al., 2009*, respectively). Of the 16 neurons described here, 75% were situated on the posterior side of the neuropil (schematized as a gray shaded area). However, soma locations varied within and across cell types (*Figure 1C*).

Many STG neurite trees contain a 'hand-like' structure, defined as a thick segment that 'sends off a number of secondary neurites' and emerges from a comparatively thin primary neurite (Example indicated in *Figure 1D* as 'H'; *Bucher et al., 2007*). This hand-like structure was observed in every GM and LG neuron, but not in any LP neurons. Of the neurons used in this study, half of the PD neurons had something that could be considered hand-like. Thus, relative to the other three neuron types, LP neurons can be distinguished by their lack of hand-like structures. We also observed the presence of 'hair-like' neurites: very thin (<1 µm) neurites that are more than 50 µm in length in all neurons.

### Expansive and complex neurite trees

Each of the neurons in *Figure 1* exhibits an expansive and complex neurite tree that ramifies throughout the STG neuropil. All 16 neurons examined had more than 10 mm of neurite in their neurite trees (*Table 1*). Despite a high mean total wiring length across all neurons examined (pooled mean ± SD = 37515 ± 27498 µm; N = 16), there was quite a bit of variability across neurons (CV = 0.73; N = 16). Across neuron types, GM neurons presented significantly greater mean total wiring lengths relative to the other three neuron types (ANOVA; [F(3,12)=8.34, p=0.0029]; Tukey HSD) with a mean ± SD of 74, 890 ± 26581 µm. LG, LP, and PD neurons had within-cell-type mean (± SD) total wirings of 30,583 ± 18,346 µm, 24,321 ± 11,777 µm, and 20,266 ± 65,805 µm, respectively – all less than 50% of the mean GM total wiring length (*Figure 2A*). GM neurons also had a greater number of soma-to-tip neurite paths (*Figure 2B*, *Table 1*; ANOVA; [F(3,12)=7.14, p=0.0052]; Tukey HSD).

We considered the complexity of these neurons' neurite trees with two approaches: Sholl analysis and spatial density analysis. Each of these analyses provides a characterization of a neuron's arborization patterns. Sholl analysis provides a measure of neurite density as a function of distance, from soma to terminating tips. Alternatively, spatial density analysis preserves the neuronal structure and provides a graphical depiction of neurite density.

In a somato-centric neuron where the neurite tree surrounds the soma, such as in starburst amacrine or stellate cells, Sholl analysis provides a characterization of neurite density as a function of distance from the soma. In this approach, concentric circles are overlaid on the image of a neuronal structure. Neurite density is characterized by counting the number of neurite intersections with increasing radius from the centroid, which is typically aligned with the neuronal soma (*Sholl, 1953*). This can be extended to three dimensions by using spheres instead of circles. Because STG neurons are unipolar rather than somato-centric, we adapted the three-dimensional Sholl analysis by linearizing the neurites and computing Sholl intersections for path length relative to the soma, rather than keeping the neurons in their innate three-dimensional conformation (*Figure 2C*). These data are

**Table 1.** Anatomical properties of GM, LG, LP, and PD neurons. Soma-to tip neurite lengths, tortuosities, and diameters are reported as a mean ± standard deviation, with the exception of the initial primary neurite (1°) diameters (for which there are single values for each neuron). Soma-to-tip neurite, branch point, and subtree values are counts for each neuron. These data are plotted in *Figures 2*, *4*, *7*, *8* and *10*.

| Cell type | Soma-to-Tip neurites | Branch points | Subtrees | Total wiring (µm) | Tortuosity | Soma-to-Tip neurite length (µm) | Neurite diameters (µm) | | | |
|---|---|---|---|---|---|---|---|---|---|---|
| | | | | | | | 1° | 2° | 3° | Tip |
| GM | 2675 | 2671 | 30 | 60,796 | 2.6 ± 2.4 | 420.7 ± 113.4 | 14.7 | 13.3 ± 10.4 | 2.4 ± 1.3 | 0.6 ± 0.3 |
| GM | 7109 | 7125 | 31 | 89,148 | 2.0 ± 1.3 | 125.9 ± 26.0 | 18 | 6.3 ± 6.7 | 2.8 ± 1.9 | 3.1 ± 3.4 |
| GM | 2676 | 2675 | 47 | 45,502 | 1.6 ± 0.7 | 501.7 ± 157.9 | 5.5 | 8.6 ± 5.1 | 4.6 ± 3.9 | 1.1 ± 0.3 |
| GM | 5985 | 5984 | 80 | 104,115 | 3.9 ± 2.1 | 664.2 ± 323.9 | 11.7 | 15.8 ± 14.1 | 17.6 ± 13.6 | 3.4 ± 1.2 |
| LG | 2003 | 2002 | 25 | 20,489 | 6.8 ± 6.4 | 399.7 ± 201.7 | 12.6 | 9.6 ± 6.4 | 2.1 ± 1.2 | 0.6 ± 0.7 |
| LG | 706 | 705 | 15 | 10,061 | 3.2 ± 1.9 | 344.5 ± 127.9 | 12.4 | 7.3 ± 5.6 | 3.0 ± 2.4 | 0.4 ± 0.2 |
| LG | 2461 | 2460 | 19 | 42,889 | 1.8 ± 1.0 | 372.9 ± 115.2 | 11 | 15.0 ± 8.6 | 3.1 ± 1.6 | 0.5 ± 0.2 |
| LG | 2310 | 2311 | 24 | 48,894 | 3.1 ± 1.5 | 371.4 ± 139.3 | 28.7 | 11.8 ± 8.2 | 3.3 ± 1.9 | 0.6 ± 0.4 |
| LP | 2016 | 2015 | 22 | 40,833 | 2.7 ± 1.8 | 402.4 ± 145.4 | 16.4 | 34.0 ± 17.0 | 7.1 ± 4.8 | 1.7 ± 1.1 |
| LP | 705 | 704 | 12 | 17,914 | 2.7 ± 1.6 | 561.1 ± 119.2 | 19 | 11.5 ± 3.4 | 7.0 ± 6.5 | 2.2 ± 1.1 |
| LP | 500 | 499 | 9 | 14,202 | 1.4 ± 0.5 | 567.5 ± 99.5 | 20.6 | 9.8 ± 5.6 | 5.0 ± 2.9 | 1.1 ± 0.9 |
| LP | 1709 | 1708 | 20 | 24,335 | 3.1 ± 2.6 | 540.8 ± 163.5 | 21.1 | 8.0 ± 6.4 | 3.9 ± 3.4 | 1.3 ± 1.3 |
| PD | 425 | 424 | 2 | 19,483 | 3.4 ± 3.1 | 414.9 ± 119.2 | 15 | 9.4 ± 4.6 | 9.3 ± 6.1 | 2.1 ± 1.9 |
| PD | 419 | 418 | 3 | 15,113 | 2.5 ± 1.2 | 328.1 ± 67.2 | 14.9 | 4.5 ± 3.3 | 9.3 ± 17.0 | 2.0 ± 1.4 |
| PD | 1598 | 1597 | 15 | 29,758 | 2.8 ± 1.7 | 329.8 ± 64.4 | 54.8 | 15.8 ± 15.7 | 8.5 ± 16.3 | 1.2 ± 0.7 |
| PD | 411 | 410 | 5 | 16,710 | 2.0 ± 1.1 | 339.7 ± 71.0 | 22.4 | 20.6 ± 15.5 | 13.1 ± 15.7 | 1.7 ± 1.0 |

shown in the violin plots in *Figure 2D*. Because somata are located at varying distances from the neuropil, the Sholl counts at each distance are normalized (described in Materials and methods). Across and within cell types, there are no apparent trends in this characterization of macroscopic branching patterns. It is interesting to note, however, that some distributions are multi-modal, suggesting that there is more branching at certain distances from the soma, within each neuron. For example, the fourth, right-most LG neuron (blue), has at least three distinct peaks in its distribution. In contrast, the first, left-most LG neuron appears to have primarily one broad peak in its distribution (*Figure 2D*).

Spatial density plots (*Figure 3*) illustrate neurite density. The skeletal reconstruction of each neuron is composed of points in the x, y, and z dimensions. For each neuron, the probability distribution function of these points was determined with Gaussian kernel density estimation. The color map represents these probabilities (between 0 and 1), which are scaled within each neuron. Based on these graphical representations, there appears to be no obvious patterning in neurite arborizations within each neuron type. Although GM neurons appear to have distinct regions of high neurite density, PD neurons are more uniformly distributed about the neuropil.

## Branching patterns

Cable theory suggests that the extent of branching across the neurite tree influences voltage signal propagation and, ultimately, a neuron's input-output function (*Rall, 1959*; *Rall and Rinzel, 1973*; *Rinzel and Rall, 1974*). Neurite arborizations and sub-trees can also reflect the connectivity of the circuit and locations of pre-synaptic inputs. GM neurons had greater branch point numbers than the other three cell types (ANOVA, [F(3,12)=7.12, p=0.0053]; Tukey HSD; *Figure 4A*; *Table 1*).

Previous work in Purkinje (*Wen and Chklovskii, 2008*) and cortical neurons (*Bielza et al., 2014*), for example, has suggested neuritic arbors presenting branch angles of <90° minimize wiring cost, given the neuron's circuit-level connectivity (*Cherniak, 1992*; *Wen and Chklovskii, 2008*). Branch angles were calculated across all branch points in each of the 16 neurons (*Figure 4B*). Branch angle

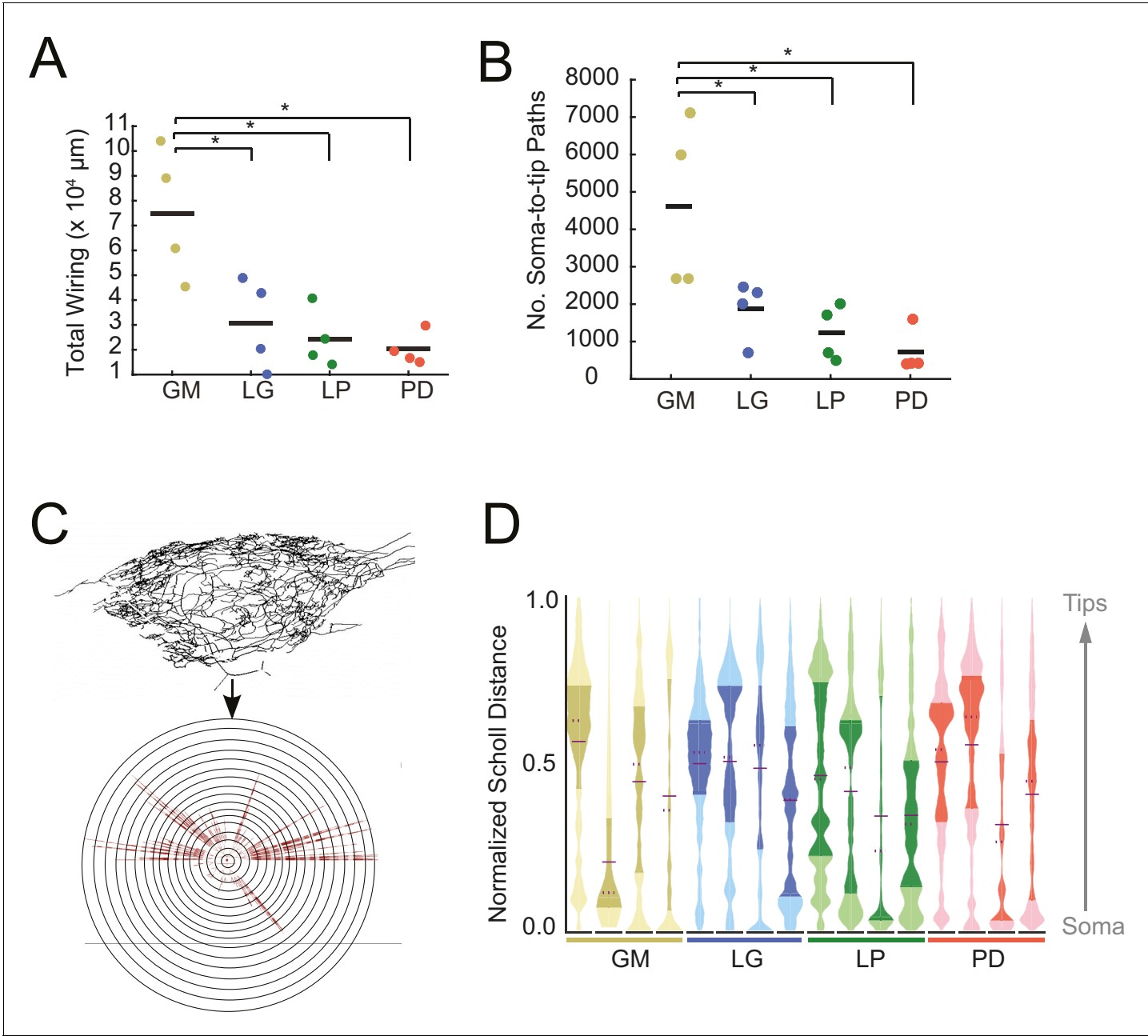

**Figure 2.** STG neurons have expansive and highly-branched neurite trees. Total wiring lengths, soma-to-tip neurite path lengths, and Sholl intersections were calculated from three-dimensional skeletal reconstructions generated in KNOSSOS from high-resolution confocal image stacks of Lucifer yellow neuronal dye-fills. (A) Total wiring, excluding axons, was greater than 10,000 μm across all neurons (N = 16; pooled mean ± SD = 37515 ± 27498 μm). GM neurons had greater total wiring than LG, LP, and PD neurons (ANOVA; [F(3,12)=8.34, p=0.0029]; Tukey HSD). (B) Mean soma-to-tip neurite paths varied across neurons but were greater in GM neurons than the other three neuron types (ANOVA; [F(3,12)=7.14, p=0.0052]; Tukey HSD). (C) For each neuron, in order to count the number of Sholl intersections as a function of distance from the soma, we first linearized the neurite paths of the three-dimensional structure (top) relative to the soma. This can be thought of as simply stretching the neurite paths radially from the soma (bottom). The number of neurites per concentric sphere intersection (or Sholl intersections) were quantified as a function of distance from the soma. (D) Sholl intersection violin plots for each neuron. Across neurons, Sholl intersections or neurite densities (counts, normalized to maximum within each neuron, on x-axis) vary with distance from the soma (y-axis, also normalized to the range of neurite path lengths within each neuron; see Materials and methods).

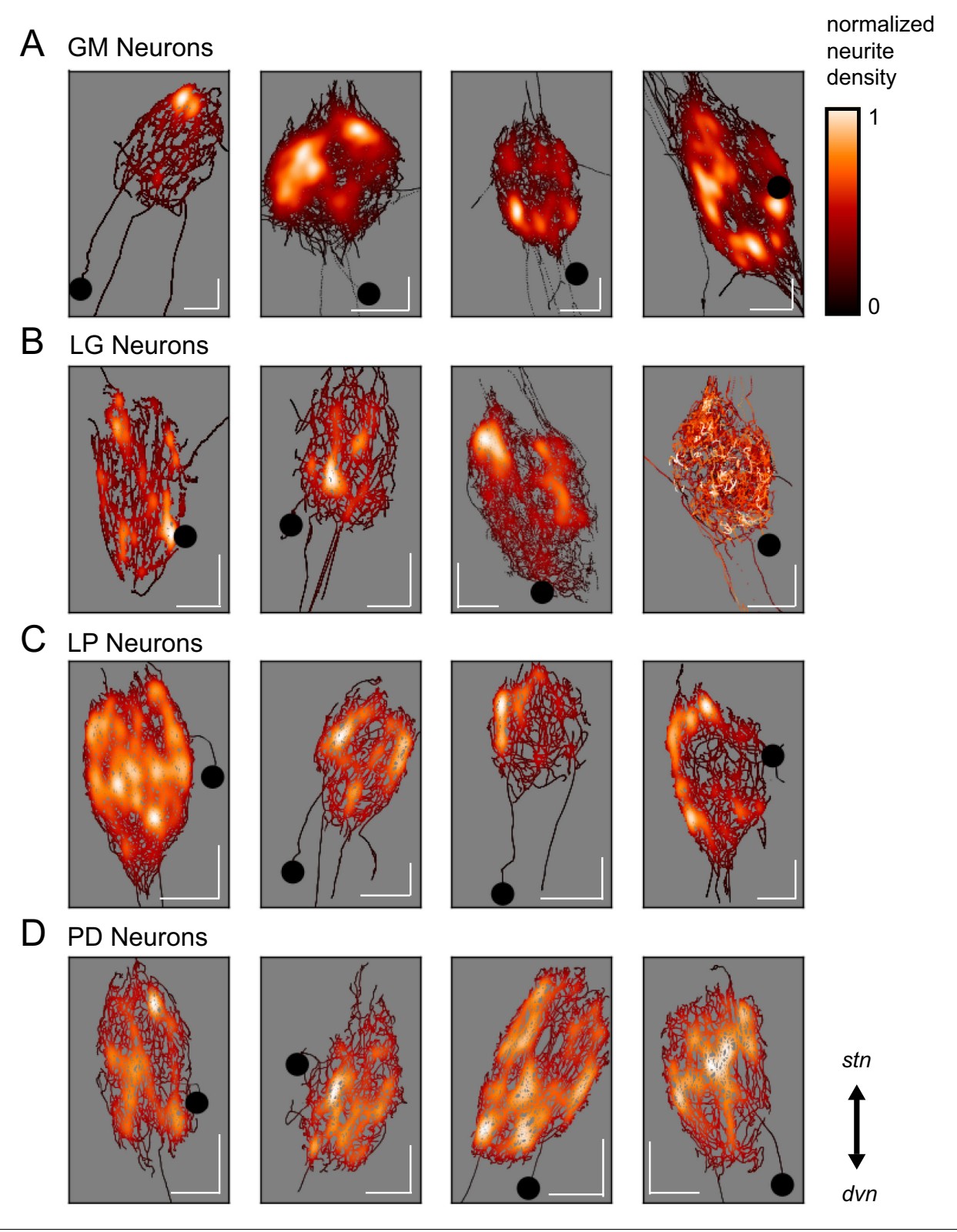

**Figure 3.** Variable spatial density profiles within and across neuron types. Spatial density profiles were generated using Gaussian kernel density estimation in the x and y dimensions. This calculation generated a probability distribution function for the points composing each skeletal reconstruction. The black-red-white color map is scaled within each neuron (see Materials and methods) and represents relative neurite density.White
*Figure 3 continued on next page*

*Figure 3 continued*

values indicate areas of high neurite density and black values indicate areas of low neurite density. White scale bars = 100 µm; black circles indicate somata locations. Each plot is oriented such that the *stn* is projecting upward and the *dvn* is projecting downward (as indicated).

distributions were similar across cell types and individual neurons (*Figure 4C*). The majority of branch angles fell between and 0 and 100 degrees. Mean GM branch angles were 70.7 ± 1.7 (median: 71.4 degrees). Similarly, LG branch angles averaged 70.6 ± 1.5 (median: 70.7 degrees), LP branch angles averaged 68.6 ± 0.6 (median: 68.8 degrees) and PD branch angles had an average value of 70.3 ± 0.8 (median: 70.0 degrees).

Previous studies characterizing neurite arborization in other neuron types, focused predominantly on bifurcating branch points (*Wen and Chklovskii, 2008*; *van Pelt and Uylings, 2011*; *Kim et al., 2012*). However, as is made evident by the presence of hand-like structures, branch points appear more complex in STG neurons. Across all neurons, we also quantified the frequency of bi-, tri-, and multi-furcating branch points (*Figure 5A–C*). *Figure 5B* shows scatter plots for each of the 16 neurons. To avoid visual bias against neurons with fewer total branch points, neurons of the same type were grouped and the proportions of bi-, tri-, and multi-furcations (quantified as the number of daughter neurites/branch point) were calculated. Bifurcations (daughter neurites/branch point = 2) were the most common across all neurons. Higher order multi-furcations were also present in each of the neurons examined here. GM neurons displayed more multi-furcations (with $\geq$4 daughter neurites) relative to other neuron types. That said, the differences in the proportions of bi-, tri-, and multi-furcations across cell types are quite small (*Figure 5C*).

The symmetry of neurite lengths following bifurcating branch points may speak to whether neurite growth occurs in a proportional manner during development. We calculated the symmetry index (SI) across all bifurcating branch points in each neuron (N = 16), which is defined as the ratio of the sums of downstream neurite lengths on either side of the bifurcation (shorter neurite length as the numerator, longer as the denominator; *Van Pelt et al., 1992*; *de Sousa et al., 2015*). Therefore, SI = 1 indicates symmetric and SI <1 indicates asymmetric neurite lengths on either side of a given bifurcation (*Figure 6A*; an example neurite path is shown in *Figure 6B*). For GM, LG, LP, and PD neurons, SI values (mean ± SD) were 0.48 ± 0.11 (median: 0.51) 0.44 ± 0.11 (median: 0.42), 0.55 ± 0.05 (median: 0.55) and 0.66 ± 0.03 (median: 0.66), respectively (*Figure 6C*). *Figure 6C* shows the full distributions of SI values for each neuron. Qualitatively, the SI distributions in PD neurons appear to be positively shifted relative to the other three cell types. Analysis of mean SI values across cell types (*Figure 6D*) reveal that PD neurons present more symmetrical neurite arborizations relative to other cell types (ANOVA; F(3,12) = 5.45, p=0.01; Tukey HSD).

## Subtrees

We quantified the number of putative subtrees in each neuron. We intuitively defined a subtree as any collection of branches projecting from the same secondary branch; thus, the number of subtrees will be equal to the number of secondary branches emerging from the main, primary neurite. Across all neurons, the number of subtrees varied between 2 and 80 (*Figure 7A*; *Table 1*). GM neurons displayed more subtrees (47 ± 23) than LP (18 ± 6) and PD (6 ± 6) neurons. However, there were no other significant distinctions between the other neuron types (*Figure 7A*; ANOVA, [F(3,12)=7.6, p=0.0041], Tukey HSD). As is clear from this scatter plot, subtree numbers are quite variable within each neuron type.

We considered the neuritic field of each neuron's subtrees to determine if they show any spatial specificity or tiling in the neuropil (*Figure 7B*). Using the three-dimensional coordinates of each subtree's terminal tips, we calculated each subtree's center of mass (consistent with analyses in *Wilensky et al., 2003*), and approximated each subtree's neuritic field by fitting an elliptical volume to the tip coordinates and centering this ellipse to the subtree's center of mass (*Figure 7B*, middle). This can be visualized when projected into the x-y plane, which represents a birds-eye view of each subtree's elliptical neuritic field (*Figure 7B*, right). *Figure 7C–F* shows the root coordinates for all the subtrees of one neuron and their corresponding neuritic fields in the x-y plane (left axes). In the

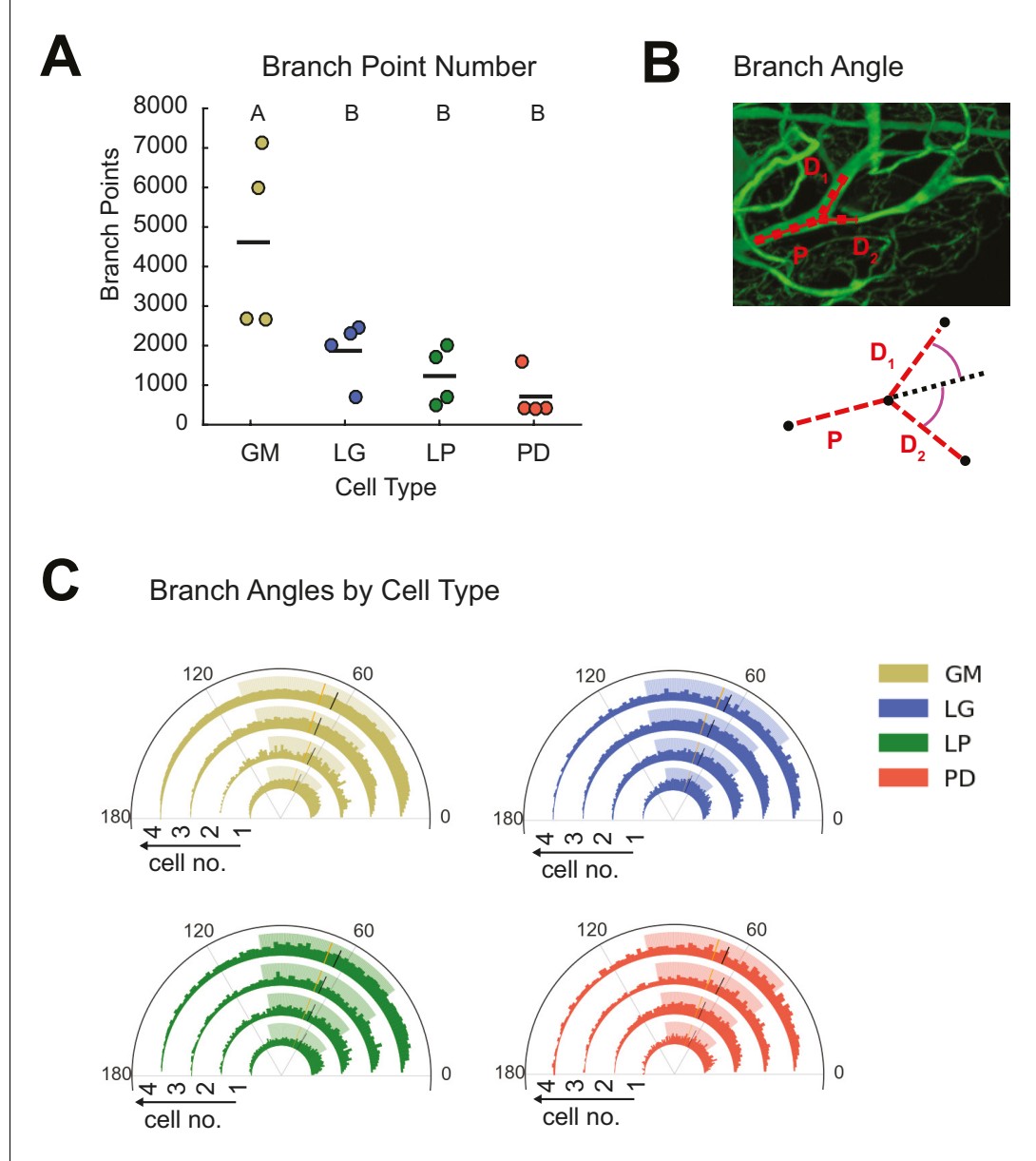

**Figure 4.** STG neurons present variable branch point numbers and angles. Branch points numbers and angles were extracted from three-dimensional skeletal reconstructions generated in KNOSSOS from high-resolution confocal image stacks of Lucifer yellow neuronal dye-fills. (**A**) Branch point numbers were highly variable across neuron types and GM neurons presented higher branch point numbers than other cell types (letters are indicative of ANOVA results; [$F_{(3,12)}$=7.12, p = 0.0053]; Tukey HSD). (**B**) Angles were calculated at bifurcating branch points as the angle (purple) between a given daughter branch ($D_1$ or $D_2$) and the hypothetical continuation of the parent (P) branch (dashed line). (**C**) Branch angle histograms show a wide range of branch point geometries within single neurons. Sets of four concentric histograms show the branch angle distributions of four neurons of a given neuron type (indicated by color in key on right). Solid lines indicate means, dashed lines indicate medians. Shaded ranges of histograms are indicative of the 25–75% confidence interval. There were no measurable statistical differences across individual neurons or neuron types).

four examples shown, the actual subtree neuritic fields appear to be compact and confined to particular areas in the x-y plane, or tiles in neuropil space.

We completed a bootstrap analysis to test if such neuropil tiling would arise from a random distribution of subtree tip coordinates. In this approach, we scrambled the observed tip coordinates and reassigned them to any subtree, while maintaining the same number of tips per subtree as observed in the actual neuronal structure. *Figure 7C–F* (right axes) shows an example of one scrambled tip

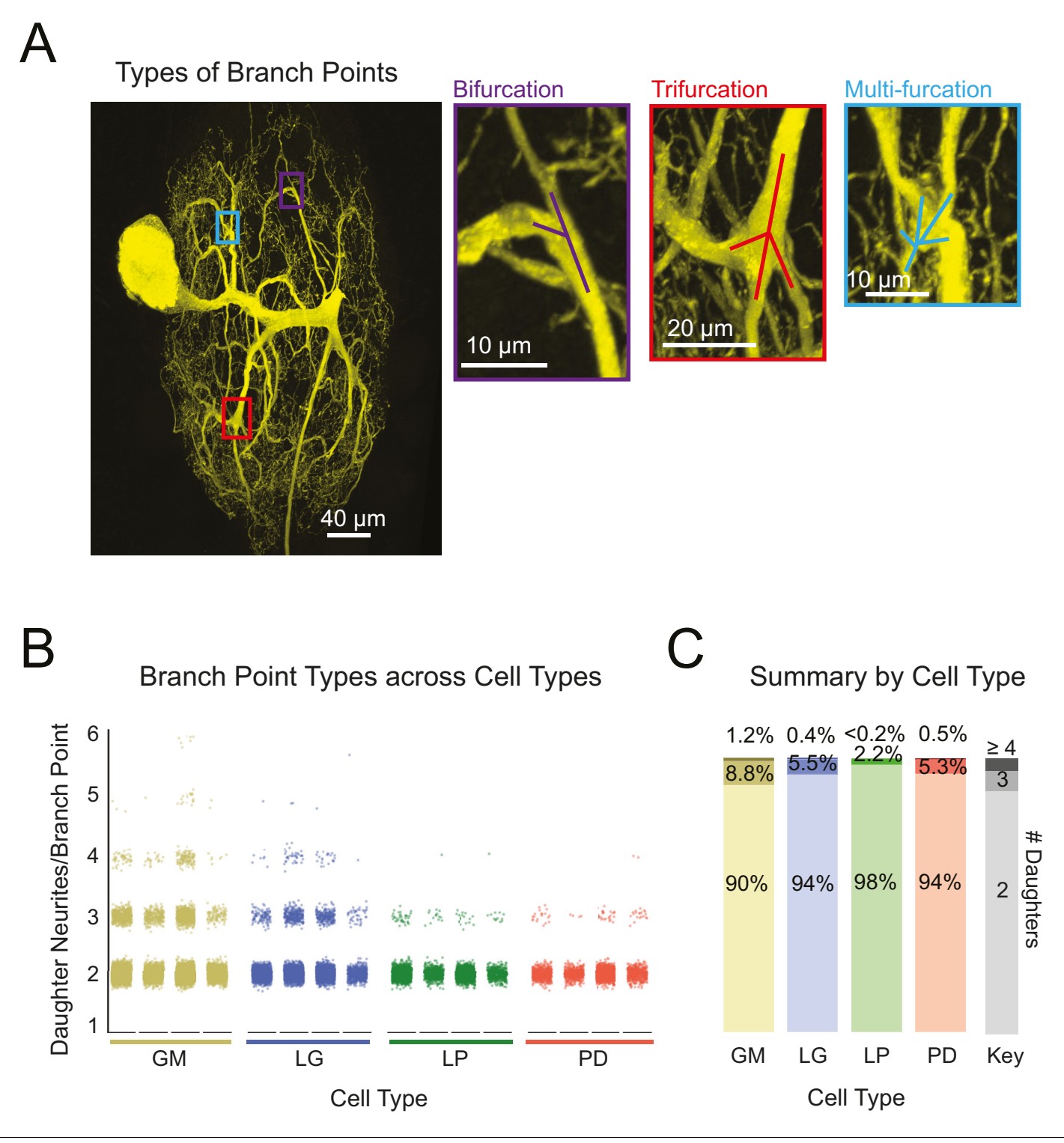

**Figure 5.** Variable branch point geometries in STG neurons. Branch point types were detected in three-dimensional skeletal reconstructions generated in KNOSSOS from high-resolution confocal image stacks of Lucifer yellow neuronal dye-fills. (A) Examples of branch point types (right) are indicated as corresponding boxes (color-coded) in a single neuronal dye-fill (left). (B) Scatter plots show the number of daughter branches per branch point (or branch point type, where 2 = bifurcation, 3 = trifurcation, and >4 indicates a multi-furcation) for every branch point in the skeletal structure of an individual neuron (n = 4 for each neuron type, indicated on x-axis). (C) Stacked bars summarize the percentage of branch points that are bifurcations (# daughter branches = 2), trifurcations (# daughter branches = 3), and multi-furcations (# daughter branches ≥ 3), as indicated by gray-shaded key (right).

*Figure 5 continued on next page*

Figure 5 continued

The landscape of branch point types was relatively similar across cell types, wherein multi-furcations accounted for ~10% or less of the branch points within each neuron. Nonetheless, unconventional tri- and multi-furcating branch points were observed in all neurons characterized in this study. Neuron types (n = 4 of each) are color-coded throughout figure: gold = GM neuron, blue = LG neuron, green = LP neuron, red = PD neuron.

iteration of each of the four example neurons. In each of the scrambled iterations, there is a notable increase in overlap between subtree neuritic fields and neuritic field radius, relative to the actual subtree distributions.

We generated 2000 scrambled iterations for each of the 16 neurons. To compare within and across neuron types, we measured two metrics that characterize the degree of overlap between subtree neuritic fields. First, we simply counted the number of overlapping subtree neuritic fields across the 16 neurons (scatter plot points) and in their corresponding scrambled-tip populations (*Figure 7G*). For comparison, histograms showing the distributions of subtree neuritic field radii across the scrambled-tip populations are overlaid on these same axes. In 13/16 neurons, the subtree neuritic fields of the actual neurons show less overlap in space than the scrambled-tip populations.

*Figure 7H* shows the mean subtree neuritic field radii for each neuron (scatter plot points) relative to those of the scrambled-tip populations (histograms). Across all neurons, the mean subtree neuritic field radii observed in the actual neuronal subtree arrangements are smaller than those of the scrambled-tip population (bootstrap analyses yielded p-values <0.01 within each neuron). Thus, the actual subtree neuritic fields are more compact relative to those generated by random tip distributions. Taken together, this analysis suggests that STG neuronal subtrees may be organized to span the neuropil in a manner that minimizes overlap between subtrees and their neuritic fields. This arrangement is unlikely to arise from random distribution of terminal tips.

## II. Fine structure

A neuron's fine geometrical properties: its neurites' lengths, tortuosities, diameters, and branching patterns, are a reflection of the distributed cable properties influencing current flow and propagation of voltage signals throughout each neuron's neurite tree (*Rall, 1977*). We found these metrics to be quite variable within and across neuron types, showing no obvious cell-type-specific rules in terms of neurite growth.

### Neurite lengths

Across all 16 neurons examined, soma-to-tip neurite lengths ranged between 5 and 1778 µm, with the majority of branches ranging between 200 and 800 µm. Distributions of neurite lengths across all 16 neurons and the four cell types are shown in *Figure 8A*. There are no evident cell-type-specific trends in soma-to-tip neurite length distributions (ANOVA; [$F_{(3,12)}=1.48$, p=0.2698]; Tukey HSD).

### Neurite diameters

We manually measured neurite diameters of primary (1°), secondary (2°), tertiary (3°), and terminating neurites (tips; schematized in *Figure 8B*). The 1° neurite refers to the neurite path extending from the soma, which is often the widest in diameter. 2° and 3° neurites branch sequentially from the 1° neurite. Grouping all neuron types together: 1°, 2°, 3°, and tips have diameters that decreased with each class, such that the mean 1° diameter across all neurons was 18.7 ± 11.1 µm, and the mean tip diameter across all neurons was 1.5 ± 1.5 µm (mean ± SD), respectively (*Figure 8C*). However, it should be noted that diameters varied within each neurite class: 1° neurites presented diameters with a CV = 0.6; 2° diameters varied with CV = 0.9; 3° diameters varied with CV = 1.4; and tip diameters varied with CV = 1.0. These same relationships can be observed within each neuron type (*Figure 8D*).

### Rall power at branch points

We examined the parent-daughter neurite radius ratios at four branch point classes: initial (first branch point relative to the soma), primary-secondary (P-S), secondary-tertiary (S-T), and terminal (at least one daughter branch is a terminating tip) bifurcating branch points (*Figure 9A*). We quantified

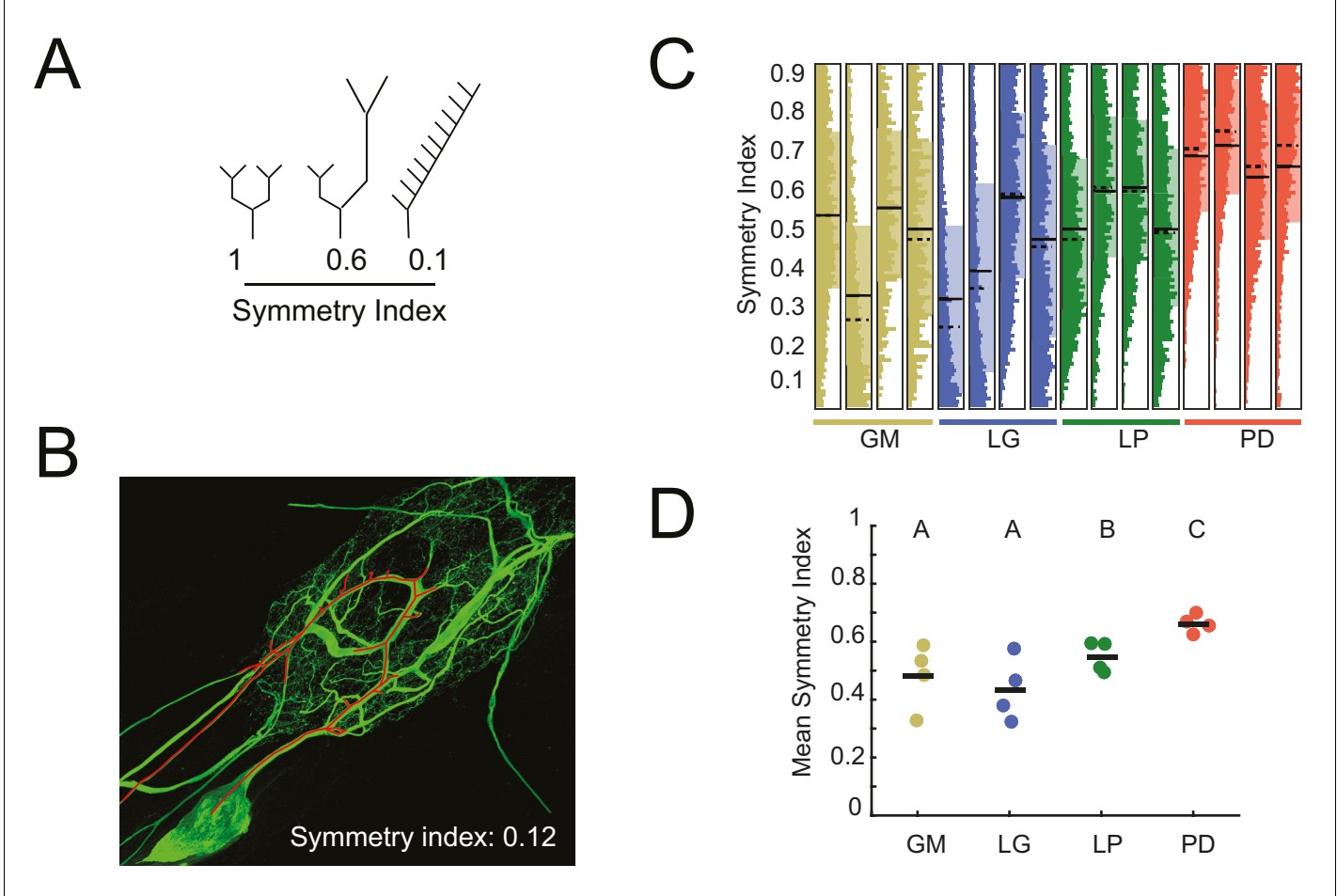

**Figure 6.** Asymmetric neurite arborizations in STG Neurons. Symmetry indices (SI) were calculated for every branch point detected in the three-dimensional skeletal reconstructions generated in KNOSSOS from high-resolution confocal image stacks of Lucifer yellow neuronal dye-fills. For a given branch point, the SI was calculated as the ratio of the sums of downstream neurite lengths on either side of the bifurcation (shorter neurite length as the numerator, longer as the denominator). (A) Schematics of branch points with varying SI; SI decreases as neurite lengths on either side of the first branch point become increasingly asymmetrical in length (from left to right). (B) Traced neurite path (red) and its corresponding SI of 0.12, which is akin to the rightmost example in A. (C) Vertical histograms show the distributions of SI values across all branches for each neuron (cell type is indicated by color and labeled on the x-axis). The x-axis scales are normalized to the maximum count within each neuron. All neurons present SI values that span the range of possibilities, between 0 and 1. (D) Mean SI values plotted by cell type. Scatter points indicate mean SI values for individual cells, black line indicates mean SI for each cell type. Letters are indicative of ANOVA results across cell types (F(3,12) = 5.45, p=0.01; Tukey HSD). Neuron types (n = 4 of each) are color-coded in C and D: gold = GM neuron, blue = LG neuron, green = LP neuron, red = PD neuron.

these parent-daughter ratios in terms of their Rall Power ($X$), which is defined as the exponent $X$ required for $\sum radius_{daughters}^{X} = radius_{parent}^{X}$ (**Rall, 1959**). $X = 3/2$ is the optimal geometrical relationship between parent and daughter neurites for continuity of current (**Rall, 1959**).

When $X > 3/2$, a given daughter neurite has a shorter radius than optimal. When $X < 3/2$, a given daughter neurite has a longer radius than optimal. These data are presented in **Figure 9B–E** and summarized in **Table 2**.

The calculated Rall powers across individual neurons, cell types, and branch point classes are quite variable and deviate from the optimal 3/2 power. Mean initial Rall powers were uniformly less than 3/2, across cell types, suggesting that the secondary branches projecting from the primary neurite tend to have large diameters relative to the primary neurite at the branch point junction. In contrast, P-S, S-T, and terminal branch points presented Rall powers that varied within and across cell types. Taken together, it does not appear that branch points present optimal parent-daughter radius ratios for current transfer, as predicted by Rall.

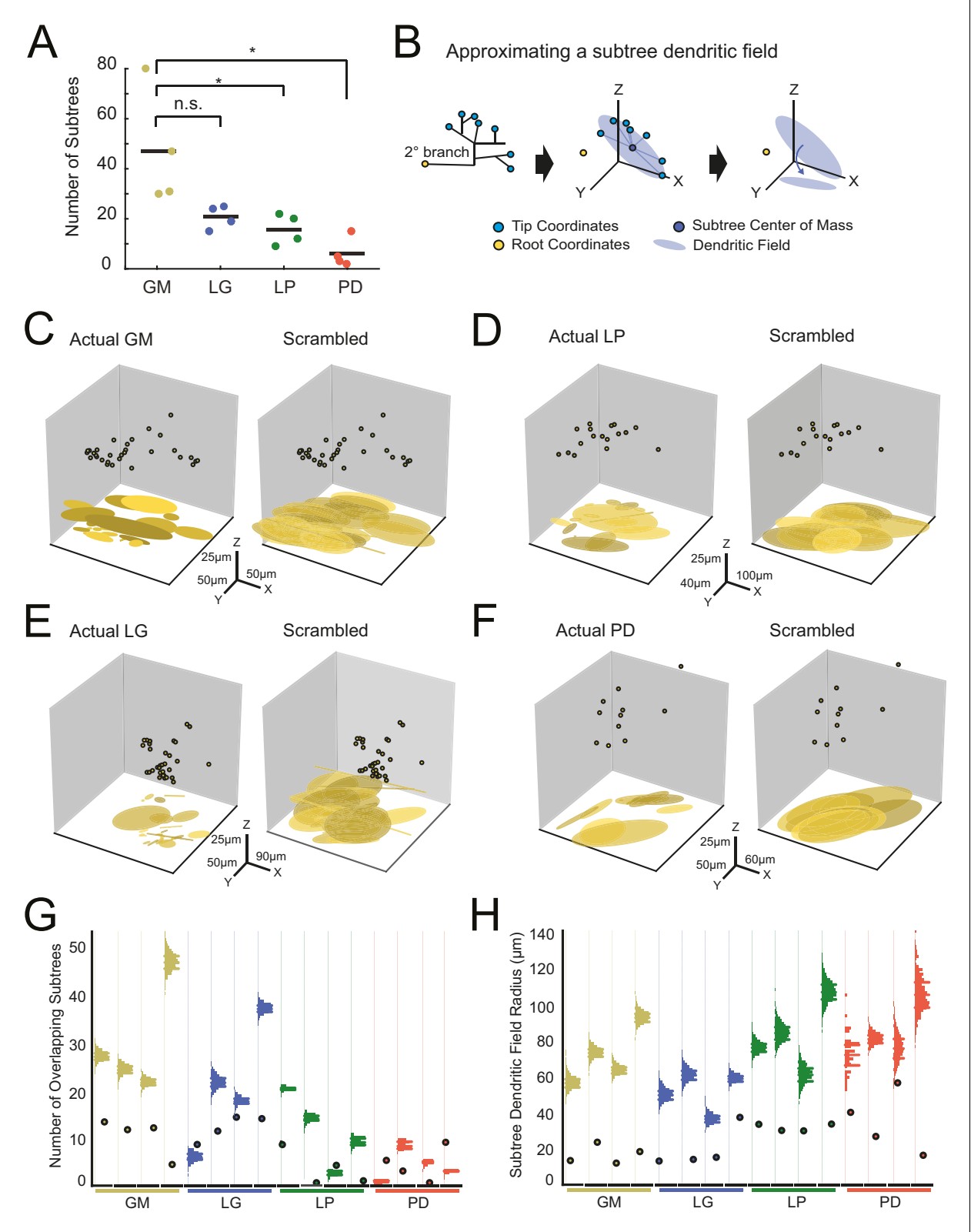

**Figure 7.** Variable subtrees that tile the neuropil. Subtrees were defined as neurites sharing a common secondary neurite exiting the main path, or primary neurite, of the neuronal structure. Thus, for each neuron, the number of subtrees was equivalent to the number of secondary neurites in the neurons corresponding to the three-dimensional skeletal reconstruction generated in KNOSSOS. (**A**) The number of subtrees varied within and across neuron types, such that GM neurons presented more subtrees than LP and PD neurons (letters are indicative of ANOVA results; [F(3,12)=7.6, p=0.0041], *Figure 7 continued on next page*

*Figure 7 continued*

Tukey HSD). (**B**) The neuritic field for a given subtree (example on left) was approximated as an elliptical volume fit (blue shaded area) and centered about the center of mass (dark blue point) of the subtree's tip coordinates (light blue points; middle). A subtree's neuritic field was visualized by projecting this elliptical volume into the x-y plane (right), which is representative of a birds-eye view of neuropil space. ( **C–F**) Subtree neuritic field plots for four representative STG neurons (left), juxtaposed to representative scrambled iterations (right). Three-dimensional scatter plots show the subtree root coordinates of individual subtrees (circular data points) and their corresponding neuritic fields projected onto the x-y plane. Left: plots display the subtree neuritic fields given the actual neuronal structure. Right: plots display the subtree neuritic fields for a representative scrambled iteration of tip coordinates for the same neuron on left, wherein tips were re-distributed into the same number of, but different subtrees, while maintaining the same root coordinates (see Materials and methods). For each neuron in **C–F** scale axes indicate the same x-y-z scale for both the actual and scrambled plots. (**G**) Scatter plots indicate the number of overlapping subtrees for each neuron (cell type indicated by color and labeled on the x-axis). Overlaid vertical histograms show the number of overlapping subtrees across the scrambled, synthetic population generated from the tip coordinates for each neuron (purple lines indicate the mean radii for each scrambled population). (**H**) Scatter plots indicate the mean subtree neuritic field radius for each neuron (cell type indicated by color and labeled on the x-axis). Overlaid vertical histograms show the range of subtree neuritic field radii across the scrambled, synthetic population generated from the tip coordinates for each neuron (purple lines indicate the mean radii for each scrambled population). Neuron types (n = 4 of each) are color-coded in A, G, and H: gold = GM neuron, blue = LG neuron, green = LP neuron, red = PD neuron.

# III. Application of wiring optimization principles

A good deal of earlier work outlines a series of metrics that speak to putative optimization principles governing neuronal morphology in developing and adult animals (*Cherniak, 1992*; *Chen et al., 2006*; *Chklovskii et al., 2002*; *Chklovskii, 2000*, *2004*; *Wen and Chklovskii, 2008*; *Kim et al., 2012*; *Rivera-Alba et al., 2014*; *Cuntz et al., 2007*, *2010*, *2012*, *2013*). Most of these wiring rules stem from Ramón y Cajal's anatomical principles of nerve cell organization, wherein he speculated that neuronal structures are optimized to conserve wiring material and conduction velocity (*Cajal, 1995*; *Cuntz et al., 2010*). In the following analyses, we have taken two approaches to quantify wiring efficiency in STG neurons.

## Tortuosity

The neuronal dye-fills in *Figure 1* illustrate the circuitous, winding nature of the neurites in each of these STG neurons. In each neuron, we compared the lengths of all soma-to-tip neurite paths to their minimal Euclidean distances. We measured this in terms of tortuosity (*Wen and Chklovskii, 2008*), which is defined as the actual neurite path length divided by the Euclidean distance (*Figure 10A*). A tortuosity of 1 is indicative of an efficient, minimal path length. A tortuosity greater than one is suggestive of a less efficient, winding path from soma to tip. Across all cell types, tortuosity distributions across all soma-to-tip neurite paths spanned beyond values of 3 (*Figure 10B*). Pooled mean tortuosities within each cell type were uniformly greater than 1, and were: 3.8 (median: 2.6), 4.7 (median: 2.9), 3.2 (median: 2.3), and 2.7 (median: 2.2) for GM, LG, LP, and PD, respectively (*Table 1*). Thus, by this metric, STG neurite paths are measurably longer than the optimal Euclidean length, given their terminal destinations. Analysis across cell types revealed no significant differences in mean tortuosities (ANOVA; [$F_{(3,12)}$: 1.2, p=0.3509]; Tukey HSD).

## Wiring cost

The tortuous neurites of STG neurons suggest that they may not adhere to widely accepted wiring optimization principles proposed in previous work on cortical and fly visual neurons (*Wen and Chklovskii, 2008*; *Cuntz et al., 2010*, *2012*). To investigate this possibility, we compared the anatomical features of these 16 STG neurons to those of synthetic, minimal spanning trees constrained by a simple wiring cost equation (as in *Cuntz et al., 2010*), using the TREES toolbox available at treestoolbox.org).

For each STG neuron, we generated corresponding minimal spanning trees (MSTs) that connect a root node to target nodes in three-dimensional space. Given the arbitrary nature of soma position in the STG (*Figure 1C*), the root node for a given MST was defined as the coordinates of the first branch point relative to the soma in the actual neuronal structure. The target nodes were randomly generated from points uniformly distributed in the three-dimensional elliptical volume approximating the space occupied by the actual neurons (*Figure 11A*). Construction of a MST connecting these nodes was constrained by a simple wiring cost equation: total cost = wiring cost + $bf$ · path length

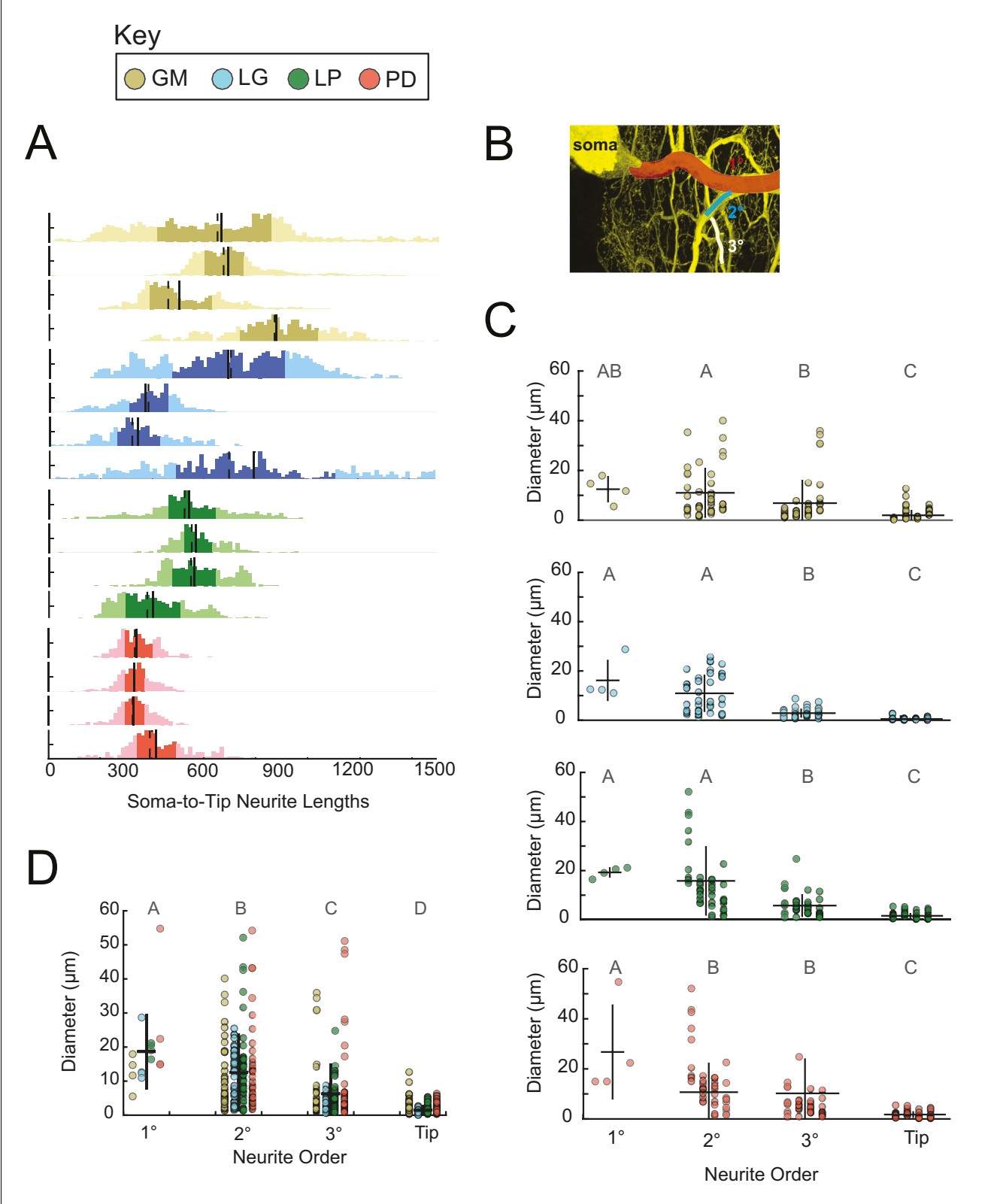

**Figure 8.** Neurite lengths and diameters. (**A**) Histograms for each neuron showing the distribution of neurite lengths across all soma-to-tip paths. Neurite path lengths were measured in three-dimensional skeletal reconstructions generated in KNOSSOS from high-resolution confocal image stacks of Lucifer yellow neuronal dye-fills. There were no cell-type-specific differences in neurite length distributions (ANOVA; [F(3,12)=1.48, p=0.2698]; Tukey HSD). Means are indicated by solid lines; medians are indicated with dashed lines; darker shaded regions indicate 25–75% confidence intervals. Neurite

*Figure 8 continued on next page*

*Figure 8 continued*

diameters were randomly sampled across primary (1°), secondary (2°), tertiary (3°), and terminating (tip) neurites and manually measured using ImageJ. (**B**) Illustration of 1° branch (red) and individual 2° (cyan) and 3° (white) branches on one neuron (z-projection of confocal micrograph at 60x magnification, neuron pseudo-colored in yellow). (**C**) Scatter plots of neurite diameters as a function of neurite order within neuron types show decreasing neurite diameters as a function of neurite order. ANOVA results are: [$F_{(3,149)}$=14.77, p<<0.001; Tukey HSD)], [$F_{(3,152)}$=67.16, p<<0.001; Tukey HSD)], [$F_{(3,152)}$=33.92, p<<0.001; Tukey HSD)], [$F_{(3,146)}$=16.43, p<<0.001; Tukey HSD)], for GM, LG, LP, and PD, respectively. (**D**) Scatter plot shows all measured diameters, across all cell types, as a function of neurite order. When all cell types are pooled are together, diameter decreases as a function of neurite order (ANOVA [$F_{(3,611)}$=89.17, p<<0.001]; Tukey HSD; indicated with letters). ANOVA analyses within each neurite order, across cell types, revealed no significant differences across cell types. For **C** and **D** pooled mean ± SD within each neurite order indicated with black lines. In **A**, **C**, and **D** neuron types (n = 4 of each) are color-coded as in key.

cost. The only variable parameter in this model is the balancing factor, *bf*, which weighs: (1) a *wiring cost* associated with the Euclidean distance between nodes and (2) a *path length cost* associated with the minimal root-to-tip neurite lengths (*Cuntz et al., 2010* describes this cost function in great detail).

*Figure 11A* shows examples of MSTs (right) generated from the root and tip coordinates of an LP neuron (left). Lower *bf* values result in MSTs with minimal total cable length at the expense of longer root-to-tip path lengths. Low-*bf* structures are consistent with relatively electrotonically compact neurons (*Cuntz et al., 2010*). Greater *bf* values result in lower root-to-tip path lengths at the expense of greater total cable length. High-*bf* structures are consistent with electrotonically compartmentalized neurons (*Cuntz et al., 2010*). It is qualitatively evident that the MSTs are not exact representations of the actual LP neuronal structure. However, if the synthetic MSTs recapitulate structural features of the actual STG, this would suggest that wiring cost is an important constraint in STG neuronal morphology. Furthermore, should a given *bf* value generate neurite trees consistent with the actual neurite trees, the *bf* value will be suggestive of the actual neuron's electrotonic structure.

*Bf* values between 0.1 and 0.85 have been used to generate realistic neuronal structures (*Cuntz et al., 2008*, *2010*, *2012*). We generated minimal spanning neurite trees with *bf* values between 0.1 and 0.6 for each of the 16 STG neurons (simulation parameters shown in *Table 3*). We then compared a number of structural features across the synthetic MSTs and actual neuronal structures. These measures include neurite lengths, tortuosities, and total wiring as described previously. We also considered branch order distributions, which provide a quantitative characterization of neuritic arborizations (*Figure 11B*, *Figure 11—figure supplements 1–4*). In brief, branch order is a property of each branch point in the neuronal structure. It is equivalent to the number of branch points between the first branch point (relative to the soma, which has a branch order of 0), and a given branch point. The branch order distributions of many of the actual STG neurons (*Figure 11B*, in black) are skewed right, with branch orders of 100 or greater, suggesting that neurite paths may branch more than 100 times before terminating. In this way, the branch order distributions reflect the complex and highly branched nature STG neurons.

*Figure 11B* shows these metrics for four STG neurons (one of each type), and their corresponding synthetic minimal spanning neurite trees. *Bf* values between 0 and 0.6 produced neurite trees with neurite length, branch order, and tortuosity distributions that span realistic ranges. Yet, no single *bf* factor collectively recapitulates each of these distributions for each neuron. For example, a minimal spanning tree with *bf* = 0 recapitulates a similar branch order distribution to that of the actual LG neuron shown in *Figure 11* (bottom). Yet, higher *bf* values are required to generate neurite length distributions similar to that of the actual LG neuron.

This discrepancy in best-fit *bf* values across morphological metrics can be observed in all 16 STG neurons (*Figure 11B*, *Figure 11—figure supplements 1–4*) and disallows a confident assignment of *bf* value within or across neuron types. Across the 16 STG neurons evaluated, tortuosity distributions were often quite broad and consistent with MSTs generated with *bf* = 0 or 0.1. Neurite distributions were less broad and consistent with MSTs generated with higher *bf* values. Branch order distributions varied widely across the 16 neurons. For many of the STG neurons, MSTs across the full range of *bf* values had total wiring lengths within 20% of the actual STG neuron (*Figure 11—figure*

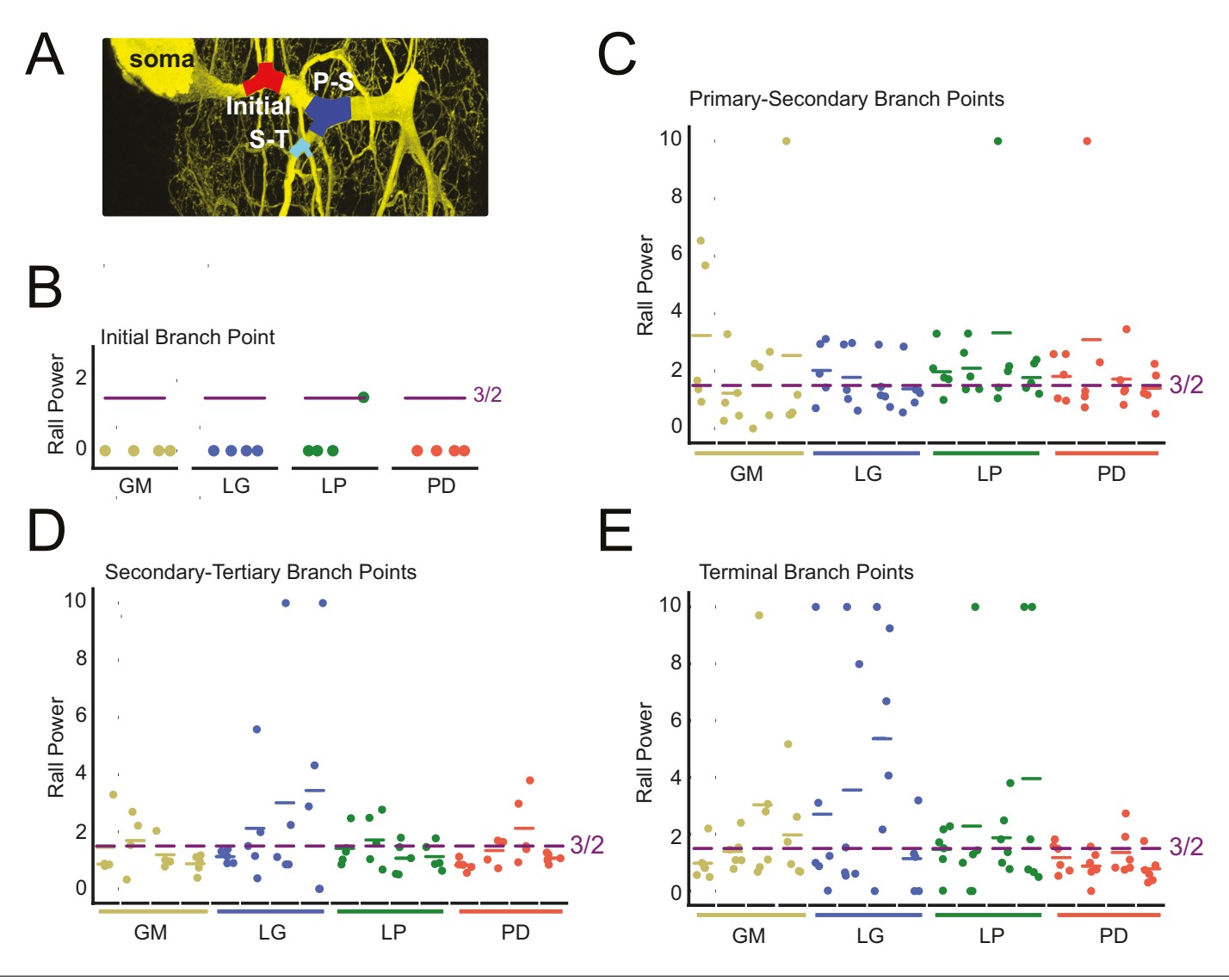

**Figure 9.** Variable Rall powers at neurite branch points. Rall powers calculated for branch points sampled across different branch orders: initial branch points (or the first branch point relative to the soma), primary-secondary (P–S) branch points, secondary-tertiary (S–T) branch points, and branch points in which at least one daughter branch was a terminating branch (terminal). Parent and daughter neurite diameters were manually measured using ImageJ, directly from the high-resolution confocal image stacks of Lucifer yellow neuronal dye-fills. (A) Examples of initial (red), P-S (blue), and S-T (cyan) branch points are shown on a maximum projection of a neuronal dye-fill. (B–E) Scatter plots show the calculated Rall powers for each neuron (as in Materials and methods and text; N = 16; cell types indicated by color and labeled on x-axis; colored horizontal bars indicate the mean Rall power for each cell) across sampled initial, P-S, S-T, and terminal branch points, respectively (n = 18–20 branch points within each branch point type). The optimal Rall power for current transfer is 3/2 (*Rall, 1959*). This is indicated on each plot with purple horizontal lines. Mean Rall powers calculated within each cell type are shown in *Table 2*. There were no statistically significant cell-type-specific differences in this metric.

supplement 1–4). However, there are several cases wherein the MSTs had total wiring lengths that deviated quite a bit from the STG neuron's total wiring length.

Taken together, this wiring cost equation alone appears insufficient to comprehensively recapitulate these structural features of STG neurons. It is plausible that STG neurite trees may simply not adhere to the wiring optimization principles enforced by this wiring cost equation.

**Table 2.** Rall powers by cell type. Rall powers were derived from neurite diameters at different branch point categories. Initial-Secondary refers to the Rall power at the first branch point relative to the soma. Terminating tips refers to branch points wherein at least one daughter branch terminates. Pooled means ± standard deviations are shown for each branch point category, for each cell type. For each cell type, pooled means ± standard deviations were calculated from six branch points in each category, for each of four neurons of that cell type. The raw data and mean Rall powers for each neuron are plotted in **Figure 9**.

| Branch point category | Rall power by cell type | | | |
|---|---|---|---|---|
| Parent-Daughter | GM | LG | LP | PD |
| INITIAL-SECONDARY | 0.001 ± 0.001 | 0.001 ± 0.001 | 0.001 ± 0.001 | 0.381 ± 0.658 |
| PRIMARY-SECONDARY | 2.132 ± 2.070 | 1.667 ± 0.857 | 4.296 ± 5.331 | 2.215 ± 1.838 |
| SECONDARY-TERTIARY | 1.305 ± 0.682 | 6.366 ± 10.098 | 1.332 ± 0.575 | 1.342 ± 0.453 |
| TERMINATING TIPS | 1.846 ± 1.449 | 6.847 ± 10.737 | 7.397 ± 10.820 | 1.049 ±. 546 |

## Discussion

Many neuron types present recognizable geometries. Yet, it remains unclear how tightly tuned or variable their macroscopic and fine structural features must be in order to generate reliable physiological output. In the present study, we carried out a detailed morphometric analysis of four STG neuron types. We found that their expansive and complex neuronal structures do not adhere to prevailing hypotheses regarding wiring optimization principles thought to govern developmental growth.

### Complex and variable morphologies

Studies in a variety of nervous systems have explored how macroscopic dendritic and axonal arborizations map to physiological function. Work in insect nervous systems showed that spatially distinct dendritic subtrees compute a weighted integration of sensory inputs (*Miller and Jacobs, 1984*; *Cuntz et al., 2008*). Studies in the mammalian retina (*Bloomfield and Miller, 1986*; *Hong et al., 2011*) and visual cortex (*Martin et al., 1983*; *Martin and Whitteridge, 1984a*, *1984b*) have demonstrated that neurons receiving sensory inputs at distinct dendritic subtrees detect various visual stimulus features. In this way, large-scale dendritic and axonal arborization patterns reflect circuit-level connectivity and contribute to input-output functions in single neurons.

Some studies (*Wang et al., 2002*; *Cuntz et al., 2008*, *2010*) have reported quantitative ranges, across many neurons of the same type, in the fine structural properties essential for current flow and integration of voltage signals arising across complex dendritic trees. The geometrical properties of any neurite path: its length, diameter, taper, and branch points, shape current flow. In this way, whether voltage events arising at disparate sites across a dendritic tree are integrated is highly dependent on neuronal geometry (*Rall, 1977*). Thus, a comprehensive understanding of how morphology maps to function requires examination of both macroscopic branching patterns and finer cable properties.

Our results suggest that STG neuronal morphologies are highly variable across animals. Additionally, our analyses did not reveal any single metric, or combination of metrics, that distinguish the four neuron types from each other. GM neurons presented significantly longer total cable lengths and branch point numbers than the other three neuron types and PD neurons showed slightly more symmetrical neurite arborizations; these are the only statistically significant cell-specific features.

Our investigation of macroscopic features such as soma position, total wiring, and branching patterns, revealed variability across neurons and within neuron types. Analysis of fine geometrical properties revealed that individual STG neurons have soma-to-tip neurite lengths that vary between 200 and 800 µm and diameters with high coefficients of variation within neurite class (primary, secondary, tertiary, or terminating neurites). Additionally, Rall powers at branch points were heterogeneous within and across neurons, suggesting variable continuity of current across the neurite tree. Given all these degrees of freedom, it is quite remarkable that these cell types show highly conserved physiological waveforms across animals.

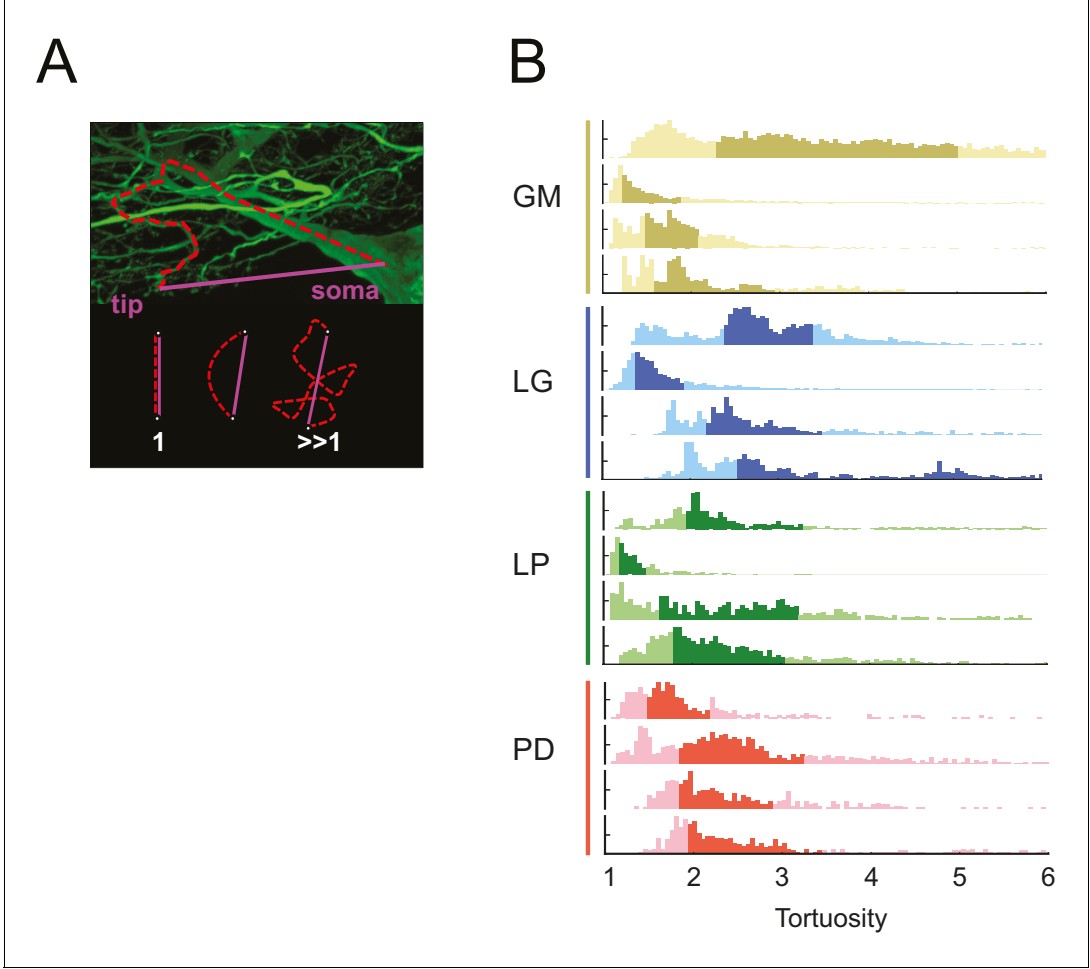

**Figure 10.** STG neurons have tortuous neurite paths. Tortuosities were calculated for every soma-to-tip neurite path detected from the three-dimensional skeletal reconstruction of each neuron (N = 16). (**A**) An example neurite path (red dashed line) from soma to terminal tip is shown on maximum projection image of an STG neuron section. The tortuosity of such a path is calculated as the ratio of the actual path length (red) over the Euclidean distance from soma to tip (purple). If the actual neurite path follows the most efficient Euclidean path, it will have a tortuosity of 1. If the neurite path deviates from this minimal path, its tortuosity will be >1. (**B**) Histograms of tortuosities for all neurite paths within each neuron (gold = GM neuron, blue = LG neuron, green = LP neuron, red = PD neuron). Darker shaded regions indicate 25–75% confidence intervals. In all cases, the distributions spanning tortuosities between 1 and 6 are shown (x-axis). In some cases, the distributions extend beyond values of 6 (not shown in this figure).

Other studies (*Wilensky et al., 2003*; *Bucher et al., 2007*; *Thuma et al., 2009*) have described neuronal morphology in the STG. However, it is important to note that these works were completed in different species with different staining procedures, lower imaging resolution, and less precise reconstruction methods. Therefore, it is not surprising that the 16 neurons in this study are more expansive and have measurably greater neurite lengths than previously reported. This is likely due to visualization of smaller processes that may have been lost in these earlier studies. Yet, many of the take-home messages are consistent with these earlier studies: soma positions of given neuron types vary across animals (*Bucher et al., 2007*; *Thuma et al., 2009*), and most neuron types are indistinguishable by soma position, maximum branch order, and total cable lengths (*Thuma et al., 2009*). While these earlier works are suggestive of morphological variability, our image resolution and morphometric analyses have allowed us to quantitatively assess this across a broader range of macroscopic and fine structural features.

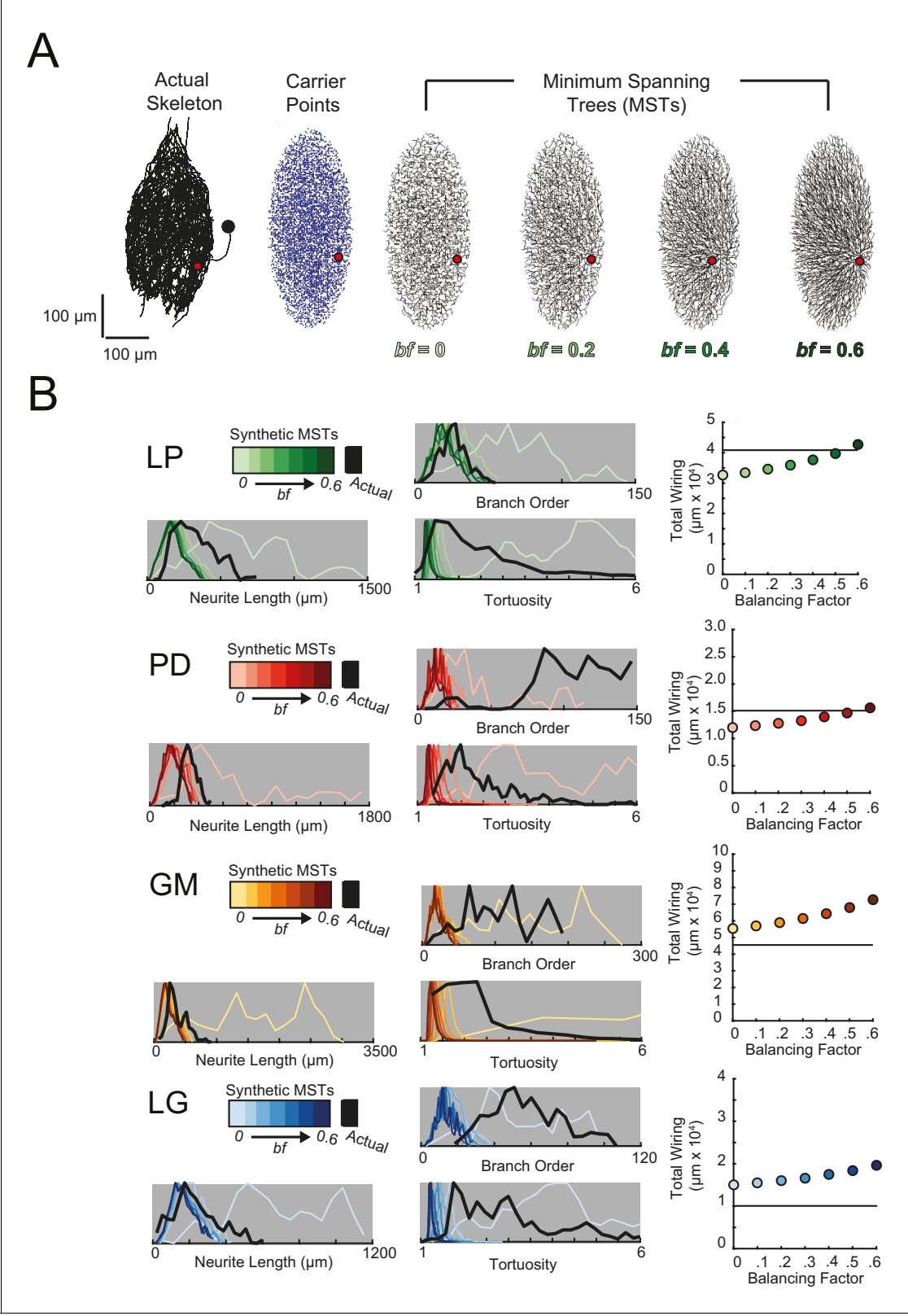

**Figure 11.** Minimal Spanning Trees (MSTs) adhering to wiring optimization rules do not recapitulate the morphological features of individual STG neurons. Synthetic MSTs adhering to the wiring cost rule: total cost = wiring cost + *bf* * path length cost (where *bf* = balancing factor) were generated using the TREEs toolbox (described in Materials and methods and by *Cuntz et al., 2010*). (**A**) An example of four synthetic minimal spanning trees (right) with varying *bf* values generated from the first branch point (location indicated as red circle on actual skeleton of an LP neuron; left) as the root. *Figure 11 continued on next page*

*Figure 11 continued*

The carrier points (blue) were randomly generated from points uniformly distributed across the elliptical volume approximating the actual volume occupied by the neuron. The number of carrier points was tuned such that the MSTs had branch point number within 20% of the actual neuron's branch point number (*Table 3*). Scale bars apply to skeleton, carrier point, and MST plots. (B) Morphological features of synthetic trees generated with *bf* values between 0.1 and 0.6 plotted against measurements from four actual STG neurons (one of each type, indicated on left and by color). Features of actual neurons are shown with black lines, whereas synthetic neurite tree data are plotted in the color scale described in the key for each neuron. Total wiring lengths, branch order distributions, neurite length distributions, and tortuosity distributions were calculated excluding axons for all synthetic and actual neurite trees. Branch order, neurite length, and tortuosity distributions were normalized to the maximum within each data set (for each *bf* value and actual neurite tree) for direct comparison of distributions despite varying neurite path counts.

The following figure supplements are available for figure 11:

**Figure supplement 1.** Minimal spanning trees (MSTs) adhering to wiring optimization rules do not recapitulate the morphological features of gastric mill (GM) neurons.

**Figure supplement 2.** Minimal spanning trees (MSTs) adhering to wiring optimization rules do not recapitulate the morphological features of lateral gastric (LG) neurons.

**Figure supplement 3.** Minimal spanning trees (MSTs) adhering to wiring optimization rules do not recapitulate the morphological features of lateral pyloric (LP) neurons.

**Figure supplement 4.** Minimal spanning trees (MSTs) adhering to wiring optimization rules do not recapitulate the morphological features of pyloric dilator (PD) neurons.

**Table 3.** Minimal spanning trees (MSTs) simulation parameters. 16 STG neuronal structures were simulated using MST analyses from the TREES toolbox (*Cuntz et al., 2010*). Carrier point numbers (column 4) were tuned to generate MSTs with branch point numbers within 20% of the branch point counts in the actual neurons (column 2, gray shaded). MSTs were generated for balancing factors between 0 and 0.6, at increments of 0.1. Columns 5–18 show branch point numbers and cable lengths for each of these MSTs. The MSTs often had cable lengths within 20% of that measured in the actual neurons (column 3), but in some cases, the MSTs were not a close approximation of the actual cable length.

| | Actual neuron | | Minimum spanning trees | | | | | | | | | | | | | | |
| | | | | Branch point numbers | | | | | | | Cable length ($\times 10^4$ µm) | | | | | | |
| Cell | Branch Points | Cable length ($\times 10^4$ µm) | Carrier Points | 0 | 0.1 | 0.2 | 0.3 | 0.4 | 0.5 | 0.6 | 0 | 0.1 | 0.2 | 0.3 | 0.4 | 0.5 | 0.6 |
|---|---|---|---|---|---|---|---|---|---|---|---|---|---|---|---|---|---|
| GM1 | 2671 | 6.1 | 8500 | 2416 | 2477 | 2512 | 2576 | 2727 | 2916 | 2971 | 16.8 | 17.2 | 17.7 | 18.2 | 19.0 | 19.9 | 20.7 |
| GM2 | 7125 | 8.9 | 23000 | 6296 | 6789 | 7030 | 7271 | 7592 | 7773 | 7994 | 2.5 | 2.6 | 2.7 | 2.8 | 2.9 | 3.1 | 3.3 |
| GM3 | 2675 | 4.6 | 8800 | 2384 | 2537 | 2665 | 2749 | 2827 | 2959 | 3026 | 5.5 | 5.7 | 5.9 | 6.1 | 6.4 | 6.8 | 7.3 |
| GM4 | 5984 | 10.4 | 19000 | 5274 | 5574 | 5762 | 5969 | 6187 | 6379 | 6571 | 14.3 | 14.7 | 15.2 | 15.8 | 16.6 | 17.5 | 18.8 |
| LG1 | 2002 | 2.0 | 6500 | 1746 | 1860 | 1976 | 2092 | 2154 | 2209 | 2302 | 1.7 | 1.8 | 1.9 | 1.9 | 2.0 | 2.1 | 2.3 |
| LG2 | 705 | 1.0 | 2300 | 641 | 672 | 696 | 710 | 750 | 768 | 815 | 1.5 | 1.5 | 1.6 | 1.7 | 1.7 | 1.8 | 2.0 |
| LG3 | 2460 | 4.3 | 8000 | 2168 | 2302 | 2389 | 2498 | 2559 | 2612 | 2707 | 4.3 | 4.4 | 4.6 | 4.8 | 5.0 | 5.3 | 5.7 |
| LG4 | 2311 | 4.9 | 7500 | 2050 | 2189 | 2282 | 2346 | 2430 | 2502 | 2603 | 2.4 | 2.4 | 2.5 | 2.6 | 2.8 | 2.9 | 3.1 |
| LP1 | 2015 | 4.1 | 6550 | 1825 | 1957 | 1998 | 2059 | 2126 | 2195 | 2263 | 3.3 | 3.3 | 3.5 | 3.6 | 3.8 | 4.0 | 4.3 |
| LP2 | 704 | 1.8 | 2330 | 649 | 666 | 706 | 716 | 748 | 773 | 788 | 2.0 | 2.1 | 2.1 | 2.2 | 2.3 | 2.5 | 2.6 |
| LP3 | 499 | 1.4 | 1705 | 452 | 493 | 509 | 534 | 531 | 560 | 574 | 1.6 | 1.7 | 1.7 | 1.8 | 1.9 | 2.0 | 2.1 |
| LP4 | 1708 | 2.4 | 5650 | 1574 | 1639 | 1697 | 1743 | 1819 | 1883 | 1921 | 3.8 | 3.9 | 4.0 | 4.2 | 4.4 | 4.6 | 5.0 |
| PD1 | 424 | 1.9 | 1400 | 370 | 398 | 422 | 444 | 442 | 464 | 501 | 1.6 | 1.7 | 1.8 | 1.8 | 1.9 | 2.0 | 2.2 |
| PD2 | 418 | 1.5 | 1400 | 383 | 406 | 426 | 425 | 455 | 460 | 477 | 1.2 | 1.2 | 1.3 | 1.3 | 1.4 | 1.5 | 1.6 |
| PD3 | 1597 | 3.0 | 5000 | 1344 | 1462 | 1519 | 1587 | 1667 | 1724 | 1776 | 3.3 | 3.4 | 3.5 | 3.7 | 3.8 | 4.1 | 4.3 |
| PD4 | 410 | 1.7 | 1300 | 363 | 389 | 397 | 396 | 429 | 442 | 451 | 1.2 | 1.3 | 1.3 | 1.4 | 1.4 | 1.5 | 1.6 |

## Sloppy geometries that are suitable for circuit function

To date, a number of experimental and theoretical works have argued that neuron growth is governed by a series of optimization principles that serve to maximize conduction velocity and minimize wiring material (*Cuntz et al., 2007*, *2008*, *2010*, *2012*, *2013*; Chklovskii 2000, *2002*; *Chen et al., 2006*; *Wen and Chklovskii, 2008*; *Rivera-Alba et al., 2014*). These studies have shown that many neuron types adhere to such optimization principles, suggesting that these are not only theoretical principles, but developmental rules that facilitate synaptic connections and circuit function.

Tortuosity measurements and comparison of STG neurons to minimal spanning trees provided within-neuron measures of wiring inefficiency. Across all 16 neurons, mean tortuosities ranged between one and seven, suggesting highly inefficient wiring. This is in contrast to other neuron types, such as mammalian pyramidal cells and GABAergic interneurons that show efficient tortuosities close to one (*Stepanyants et al., 2004*; *Wen et al., 2009*). Likewise, the neurons of the nematode *Caenorhabditis elegans* present a highly stereotyped and efficient connectome (*Chen et al., 2006*).

This sloppy morphological tuning of STG neurons is notable, but it is not the first exception to wiring optimality shown in the literature. For example, anatomical studies of motor axons at the interscutularis neuromuscular junction in mouse show a great deal of variability in branching topology of axonal arbors, as well as suboptimal wiring lengths (*Lu et al., 2009*). As suggested by *Lu et al. (2009)*, such case studies do not imply that wiring optimization principles are not at play, but may simply suggest that other factors constrain morphology in these nervous systems.

STG circuit structure and anatomy are evidently resilient to this loose morphological tuning. STG circuit function relies predominantly on slow oscillations (*Graubard and Ross, 1985*; *Ross and Graubard, 1989*) and graded inhibitory transmission (*Eisen and Marder, 1982*; *Marder and Eisen, 1984*; *Maynard and Walton, 1975*; *Graubard et al., 1980*; *Manor et al., 1997*, *1999*; *Bose et al., 2014*). Slow synaptic events undergo minimal electrotonic decrement and are resilient to the variable, complex, and extended neurite trees of STG neurons (*Otopalik et al., 2017*). This would effectively mask the physiological consequences of animal-to-animal variability in morphology.

Additionally, early electron microscopy studies describe the sparse distribution and tight apposition of synaptic input and outputs across the finer processes of the neurite tree, such that nearly every secondary process has both pre- and post-synaptic regions (*King, 1976a*, *1976b*). Further, the connections between particular pairs of neurons are sparsely distributed over multiple neurites of their neurite trees (*King, 1976a*). King suggests that this sparse distribution of synaptic contacts on finer processes gives rise to some degree of synaptic democracy, wherein no one synaptic partner "exerts an overriding influence on the neuron" (*King, 1976a*). Such distribution of synaptic connections likely contributes to the masking of morphological variability at the physiological level. Slow postsynaptic voltage events arising across sparsely distributed sites may influence transmission globally or locally. Sparse, slow voltage events, resilient to electrotonic decrement, may have the capacity to sum and alter the net physiology of the neuron. Alternatively, small postsynaptic potentials may locally influence transmission at nearby postsynaptic sites on the same branch, influencing particular synaptic connections in the circuit. Regardless, the electrotonic compact nature of these neurons, along with this tight pairing of pre- and post-synaptic sites across the neurite tree, may effectively compensate for the complex and variable morphologies observed across animals.

Theoretical studies have described the direct consequences of neuronal morphology and neuronal firing patterns (*Mainen and Sejnowski, 1996*) and shed light on how neuronal structures may be optimized for Hebbian or spike-timing-dependent computations at faster time scales (*Stiefel and Sejnowski, 2007*). We suspect that minimization of wiring and morphological precision may be more critical when synaptic and circuit-level computations occur at faster time-scales. Yet, a recent study demonstrated a great deal of variability in dendritic morphology within one class of mammalian neocortical pyramidal neurons (*Hamada et al., 2016*). Interestingly, the distance between the soma and spike initiation zone co-varies with and compensates for variability in the dendritic tree. In this case, the pyramidal neuron achieves its target physiological properties by tuning ion channel distributions to the neuron's geometry.

If developmental growth of STG neurons is not governed by the same wiring principles suggested in other neuron types, what constrains their growth? Consideration of the spatial relationships between subtrees within individual STG neurons suggests that subtrees grow to fill and tile distinct

spatial fields in the neuropil. We speculate that this may be for the purpose of maximizing surface area for the reception of the numerous neuromodulatory substances that are diffusely released in the hemolymph and by the descending inputs of the stomatogastric nerve (*Marder and Bucher, 2007*; *Blitz and Nusbaum, 2011*). Thus, in development, space-filling of the neuropil may outweigh the wiring economy that might be expected if decrement of synaptic events were critical to circuit function.

## Materials and methods

### Animals and dissections

Adult male Jonah crabs (*Cancer borealis;* carapace lengths between 13 and 17 cm) were obtained from Commercial Lobster (Boston, MA). All animals were kept in artificial seawater tanks at 10–13°C on a 12 hr light/12 hr dark cycle without food. Prior to dissection, crabs were anesthetized on ice for 30 min. Dissections of the stomatogastric nervous system (STNS) were performed as previously described (*Gutierrez and Grashow, 2009*) in saline solution (440 mM NaCl, 11 mM KCl, 26 mM $MgCl_2$, 13 mM $CaCl_2$, 11 mM Trizma base, 5 mM maleic acid, pH 7.4–7.5). In brief: the stomach was dissected from the animal. The STNS was isolated from the stomach, including: the two bilateral commissural ganglia, esophageal ganglion, and STG, as well as the lateral ventral nerve (*lvn)*, medial ventral nerve (*mvn)*, lateral gastric nerve (*lgn)*, and dorsal ganglion nerve (*dgn)*. The STNS was pinned down in a Sylgard-coated petri dish (10 ml) and continuously superfused with chilled saline.

### Cell identification

A schematic of the full experimental workflow can be found in *Figure 12*. The STG was desheathed and intracellular recordings from the somata were made with 12–40 MΩ glass microelectrodes filled with 0.6M $KSO_4$ and 20 mM KCl, amplified with 1x HS headstages and Axoclamp 2A and 2B amplifiers (Molecular Devices). For extracellular nerve recordings, Vaseline wells were built around the *lvn, mvn, lgn,* and *dgn* and stainless steel pin electrodes were used to monitor extracellular nerve activity. GM, LG, LP, and PD neurons were unambiguously identified by matching their intracellular activity with spike units on nerves containing their axons: *dgn* for GM, *lgn* for LG and the *lvn* for LP and PD.

### Neuronal dye-fills

Following electrophysiological cell identification, somata were impaled with low-resistance glass electrodes (3–16 MΩ) backfilled with 2% Lucifer Yellow CH dipotassium salt (LY; Sigma, catalog number L0144) in filtered water. Lucifer Yellow was injected into somata for 20–50 min with negative pulses of between –3 and –7 nA of 0.5 s duration at 0.1–1 Hz. The dye-fill was performed until the fine neuropil processes of the cell were visible with a fluorescent microscope (Leica MF165 FC). Preparations were fixed with 3.5% paraformaldehyde in phosphate-buffered saline (PBS; 440 mM NaCl, 11 mM KCl, 10 mM $Na_2HPO_4$, 2 mM $KH_2PO_4$, pH 7.4) for 40–90 min at room temperature within 3 hr after LY dye-fill. Preparations were washed with 0.01M PBS-T (0.1–0.3% Triton X-100 in PBS) and stored for 0–7 days at 4°C before processing.

### Dye-fill amplification and immunohistochemistry

The LY signal was amplified by addition of a polyclonal rabbit anti-LY antibody (1:500; Molecular Probes, 16 hr), followed by secondary detection with a polyclonal Alexa Fluor 488-conjugated goat anti-rabbit antibody (IgG H+L chains, highly cross-absorbed; Molecular Probes1:500; 2 hr). Preparations were then washed 4 × 15 min in filtered PBS and mounted on pre-cleaned slides (25x75 × 1 mm, superfrost, VWR) in Vectashield (Vector Laboratories, Burlingame, CA), with 9 mm diameter, 0.12 mm depth silicone seal spacers (Electron Microscopy Sciences, Hatfield, PA) under #1.5 coverslips (Fisher Scientific).

### Confocal imaging and processing

Confocal image stacks of the neuronal dye-fills were acquired with a SP5 CLEM microscope using Leica Application Suite Advanced Fluorescence (LAS AF) software. Confocal images were acquired as tiles in the x-y plane with at 63x magnification with a glycerol objective (Leica HCX PL APO 63x/

1.3 GLYC CORR CS (21°C)) at either 2048 x 2048 or 1024 × 1024 resolution. Stacks in the z-dimension were acquired in steps between 0.12 to 1.01 μm steps. These tile stacks were aligned and stitched with a GUI-based MATLAB tool (*Goeritz et al., 2013*) available at https://github.com/marderlabConfocal_Stitching.

Maximum projection images were generated from these confocal stacks using Imaris 7.0–8.1 software (Bitplane). These were filtered and down-sampled to a third of the resolution in the x and y dimensions. Imaris Slice and Surpass modules were used for adjusting contrast and brightness and to display stacks as maximum-intensity projections or as blend mode projections. Contrast and exposure were adjusted to provide maximum-intensity images.

### Three-dimensional skeletal reconstruction

Neuronal dye-fills were traced and skeletal three-dimensional skeletal reconstructions were generated using one of three software packages: Amira (FEI Visualization, Hillsboro, OR), Imaris (Bitplane), or KNOSSOS (Max Planck Institute, Heidelberg, Germany, http://www.knossostool.org). These techniques range from semi-automated (Amira) to manual (Imaris, KNOSSOS), and are reviewed elsewhere (*Donohue and Ascoli, 2011*). No major tracing differences were found between these methods in generating skeletal structures, although KNOSSOS was the fastest method.

In brief, a team of tracers generated skeletal reconstructions of neurite structures. They were instructed to place nodes at the thickest or most intense part of each neurite and to minimize the number of nodes while capturing as much curvature as possible. Tracers did not measure neurite diameters or volumes. In some cases, anastomoses appeared to be present but tracers were instructed to avoid tracing loops. The skeletons from Amira and Imaris were exported directly as *hoc* (extendable high-order calculator) files consistent with the NEURON simulation environment (*Hines and Carnevale, 2001*; *Hines et al., 2007*). KNOSSOS files were exported as *xml* collections

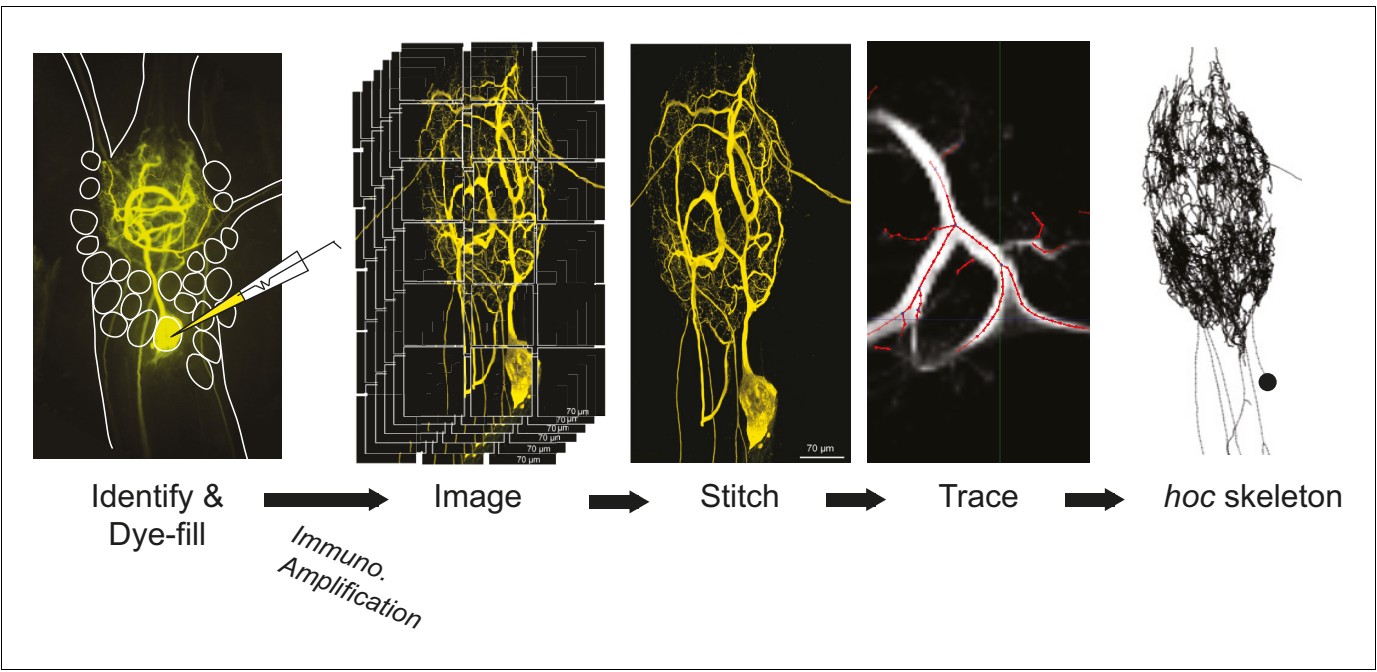

Identify & Dye-fill → *Immuno. Amplification* → Image → Stitch → Trace → *hoc* skeleton

**Figure 12.** Procedural workflow from dye-fill to morphological analysis. Neurons were identified physiologically and then filled with Lucifer yellow. Following fixation and immunohistochemical amplification with a rabbit anti-Lucifer yellow primary antibody and an Alexa Fluor 488-tagged secondary antibody, each dye-fill was imaged at 63x magnification as a series of z-stacks that tiled the neuronal structure across the x-y plane (z-stacks contained slices at increments of ~0.5 μm). These z-stacks were stitched together with custom software written in MATLAB. The image stack of each full three-dimensional neuronal structure was then manually traced (image shown is a screenshot from the KNOSSOS platform). The three-dimensional skeletal structures were then converted to *hoc* files that could be analyzed with a custom quantitative morphology toolbox composed in Python. All conversion and analytical scripts are freely available on the Marder Lab Github repository.

of edges and nodes and were adapted into *hoc* (extendable high-order calculator) files by custom Python software, which is available at: https://github.com/marderlab/Quantifying_Morphology.

## Geometry objects

All quantitative analyses were built in Python using custom geometry objects generated from the *hoc* skeletons for each neuron. These geometry objects had the following features: terminating tips (non-soma, non-axonal terminal filaments), somata (manually tagged), paths (filaments connecting somata to tips), segments (any tracer-defined collection of nodes), branches (a collection of segments spanning branch points or multi-furcations), and axons (manually selected terminal filaments that do not branch for at least 20 μm; see *Figure 12* for an example). Most of the properties examined in this study can be readily extracted from these geometry object attributes. Skeletal representations were generated by scatter plotting a subset of the geometry object's nodes. This and all code is available for download at github.org/marderlab/Quantifying_Morphology.

## Path length and tortuosity

Path lengths were calculated as the sum of the distances between the points that constitute a given branch. Geometry objects contain instances of branches and segments, each with a length attribute. The length of a path is simply the sum of the lengths of its constituent segments. Horizontal histograms were used to show the relative distributions underlying the data; means are indicated with solid black lines, medians with dashed lines (solid orange lines for radial histograms), and interquartile ranges with shaded regions. Path tortuosity is defined as path length divided by the Euclidean distance from the first node to the last node (the soma to a tip). A tortuosity of 1.0 indicates a perfectly straight path while values larger than 1.0 indicate a wandering or tortuous path.

## Branch angle

To compute branch angles, all the branch points and segments that contributed to the bifurcation were collected; in the case of a multi-furcation, the procedure was repeated with all possible combinations of branches. The endpoints of the segments were collected into groups containing a midpoint ($P_{mid}$, which is also the proximal endpoint of the parent and the daughter) and the distal endpoints of the parent ($P_1$) and daughter ($P_2$). Using the other two points as nodes of a triangle, branch angle is given by

$$\theta_{mid} = cos^{-1}\left(\frac{P_{mid}^2 - P_1^2 - P_2^2}{2P_1 P_2}\right)$$

where $P_x =$ magnitude of vector formed from points other than point *x*. Angles were reported in degrees and were taken as 180 - $\theta_{mid}$ to reference the angle against the continuation of the initial segment. Branch angles and torques were plotted on circular histograms with inter-quartile ranges indicated by shading; means and medians are indicated by the black and orange lines, respectively. These circular histograms were inspired by *Bielza et al. (2014)*, although the code is original.

## Symmetry index

Length asymmetry is found by summing the lengths of all downstream segments originating from daughter segments of a bifurcation. Symmetry index measures this property by dividing the smaller summed length by the larger summed length. Formally, the symmetry index *S* is defined as

$$S = \frac{min\{d_1, d_2\}}{max\{d_1, d_2\}}$$

where $d_1$ and $d_2$ represent the sum of the *lengths* of all downstream segments for the two daughter branches of a bifurcation. Downstream branches are all the neighboring branches arising from the daughter and any of its subsequent daughters. Intuitively, a symmetry index of 1.0 indicates a perfectly symmetrical bifurcation in terms of downstream lengths, whereas values closer to 0 indicate highly asymmetrical bifurcations.

## Branch diameter

Manual measurements were made from 2D slices of the neuron to approximate soma and branch diameters using the measure tool in the ImageJ variant FIJI (www.fiji.sc). Most STG neurites, especially large-diameter neurites, are slightly flat (compressed in the z-dimension), rather than cylindrical. Therefore, the values given here should represent a fair assessment of the parent-daughter ratios (used to calculated Rall Power) of the neurons. The diameters of the soma, initial primary neurite leaving the soma (1°), secondary branches (2°), tertiary branches (3°), and branches where one of the daughters is a terminating tip ('tip') were measured.

## Soma position

Using the maximum projections of each image stack, the coordinates for the neuropil center and soma positions were measured in pixels and converted to microns using FIJI. These were plotted on the same set of axes and translated such that the neuropil center within each preparation was situated at the origin.

## Rall power

Rall power (*Rall, 1959*) is calculated as $parent^X = daughter_1^X + daughter_2^X$, where the parent is the radius of the parent branch and daughter is the radius of the daughter neurites. Rall power values have been found to range from about 1.5 (*Rall, 1959*) to 3 (*Chklovskii and Stepanyants, 2003*). We calculated the Rall power of STG branches numerically by minimizing $parent^X - \left(daughter_1^X + daughter_2^X\right)$ and varying X.

## Sholl analysis

Sholl analysis typically superimposes concentric circles or spheres, centered at the soma, to quantify the spatial density of neurites by measuring the number of intersections of circles/spheres with neurite branches. Because STG somata are segregated from the neuropil and the neurite field, this approach was not logically consistent with the goal of Sholl analyses. Instead, we calculated an equivalent somatofugal Sholl distance by creating linearly spaced distances ranging from zero to the maximum path length and asked for each pair of nodes whether the traversing line crossed this distance. In this way, path lengths are the distances being measured and branching patterns can be elucidated. Equivalently, one could re-plot an STG neuron with the soma at the center of a 2D Sholl graph with filaments that conserve segment length branching out radially from the soma.

## Spatial density analysis

Spatial density analysis was used to visualize neurite density of individual neurons in the x-y plane. Geometry objects were generated from the *hoc* skeletons for each neuron using Python. A Gaussian kernel density estimation (from the SciPy statistics module, utilizing Scott's Rule for bandwidth selection) was used to generate a probability distribution function for all points in the x and y dimensions composing each neuron's geometry object. This resulted in a probability value (between 0 and 1) for each point that indicates its relative number of nearest neighbors. For each neuron, the geometry object points were plotted in the x-y plane and colored by the points' probabilities. The color map was scaled to the range of probabilities within each neuron. In this way, the spatial density plots can be interpreted as visualizations of neurite density. This approach was adapted from http://stackoverflow.com/a/20107592 and can be found on the Marder Lab Github.

## Subtrees

The main path was defined as the segments connecting the soma to the axon(s). LP, PD, and LG neurons have only one axon (thus, only one main path), whereas GM neurons have 3–5 axons and, therefore, 3–5 main paths. Subtrees were defined by: (1) having a root segment that departed from the primary neurite and (2) containing at least one terminating tip segment. The terminating tips of a subtree form a tip cluster. The center of mass of a subtree was calculated as the mean x-y-z coordinates of its terminating tips. The radius of a subtree tip cluster was calculated as the mean distance from the center of mass coordinates to each tip's coordinates. We used a design-based inference sampling strategy (*Geuna, 2000*) to randomly assign tips to any cluster, while maintaining the number of tips per cluster. The reassigned tips for a given subtree could originate from any subtree,

consistent with an earlier study (*Wilensky et al., 2003*). We performed this randomization between 100 and 2000 times, depending on the number of possible tip configurations and subtrees. The subtree tip cluster radii and the number of overlapping subtree tip clusters from the randomized subtrees were compared to those of the original neuronal structure.

## General and statistical analysis of skeletal reconstructions

All analyses were carried out in R 3.2.2 or Python 3.4.2 with the assistance of open-source packages, chiefly NumPy and SciPy (*van der Walt et al., 2011*), NetworkX (*Hagberg et al., 2008*), and generally used interactively with IPython (*Perez and Granger, 2007*). Although the *hoc* data files are not large, some analyses required intensive computation. The computer used was a 2014 Dell Precision T3600 with 64 GB RAM and six i7 processors running Ubuntu 14.04. A similar computer was used for the image analysis and manual tracing. The interested user will find sample *hoc* data and all required programs on the Marder Lab GitHub site (http://github.com/marderlab/Quantifying_Morphology). Users are welcome to copy or adapt the analytic and plotting tools found there with attribution. A complete *hoc* dataset can be found in the Source Data file. One-way analysis of variance or Kruskal-Wallis (when homoscedasticity is not satisfied and data are not normally distributed) were performed for statistical comparisons between cell types and Tukey honest-significant difference was (HSD) used for post-hoc pairwise comparisons in all cases. These statistical analyses are reported in the text.

## Graphical presentation

Neuron images were captured with Imaris or Amira. Plots were made in Python with Matplotlib (*Hunter, 2007*) or MATLAB (MathWorks, Natick, MA). Plotting software is available on the Marder Lab Github repositories. Figures were assembled in InkScape (http://www.inkscape.org/) or Illustrator (Adobe).

## Minimal spanning neurite trees (MSTs)

Synthetic neurite trees were generated using the TREES toolbox developed by Hermann Cuntz (Ernst Strüngmann Institute, Frankfurt, Germany) and the Häusser Laboratory (University College London, UK). This MATLAB (Mathworks, Natick, MA) toolbox is freely available with an extensive user's manual online: www.treestoolbox.org. The minimal spanning neurite trees were generated with similar methods as described in *Cuntz et al. (2010)*. For each of the 16 STG neurons, synthetic trees were constructed to connect the root node (defined by the coordinates of the first branch point relative to the soma, in the actual neuronal structure) to the target nodes. Target nodes were randomly generated from points uniformly distributed in an elliptical volume approximating the space occupied by the actual neuronal structure. The number of target nodes was tuned such that branch point numbers of the MSTs were within 0–20% of the actual STG neuron. The lengths and paths of neurite segments connecting these nodes were constrained by the simple wiring cost equation: total cost = wiring cost + *bf* * path length cost (as described in *Cuntz et al., 2010*). The balancing factor (bf) is the only variable parameter in this model and weighs the cost of minimizing total cable length versus minimizing soma-to-tip path lengths. Seven corresponding synthetic trees were generated for each of the 16 STG neurons in the present study with a range of balancing factor values between 0.1 and 0.6 (at increments of bf = 0.1). Total wiring, branch point numbers, branch order distributions, neurite length distributions, and tortuosity distributions of the synthetic trees were plotted in comparison to those of the actual skeletal structures using MATLAB. This range of bf values was chosen because the resulting MSTs recapitulated realistic ranges in these morphological features that spanned those of the actual neurons. All data and structures pertaining to the actual and synthetic neurite trees used for this simulation are freely available on the Marder Lab Github repository: http://github.com/marderlab.

## Data accessibility

Raw confocal image stacks are over 100 GB but are available upon request. Skeletal reconstructions of these neuronal dye-fills are available as *hoc* files on the neuron morphology database Neuromorpho.org under *Cancer borealis.* Morphometric data sets, analytical scripts, and minimal spanning tree simulation data are available at github.org/marderlab.

## Acknowledgements

We thank: Edward Dougherty and the Confocal Imaging Lab at Brandeis University; Richard Ho, Isabelle Moore, Dahlia Kushinsky, Lily He, and Matthew Stenerson for manual tracing of neuronal dyefills. The order among co-first authors was established by randomly drawing colored jellybeans from a large cup.

## Additional information

### Competing interests

EM: Deputy Editor *eLife*. The other authors declare that no competing interests exist.

### Funding

| Funder | Grant reference number | Author |
|---|---|---|
| National Institute of Neurological Disorders and Stroke | R37NS17813 | Eve Marder |
| National Institute of Neurological Disorders and Stroke | F31NS092126 | Adriane G Otopalik |

The funders had no role in study design, data collection and interpretation, or the decision to submit the work for publication.

### Author contributions

AGO, Data curation, Formal analysis, Funding acquisition, Validation, Investigation, Visualization, Methodology, Writing—original draft, Writing—review and editing, Wrote software for quantitative morphology analyses, Generated morphometric data sets, Performed analyses, Produced figures, Executed the wiring simulations, Wrote the original manuscript; MLG, Conceptualization, Data curation, Methodology, Conceived of and designed the study, Acquired and processed the confocal image stacks; ACS, Conceptualization, Data curation, Software, Formal analysis, Methodology, Writing—original draft, Writing—review and editing, Conceived of and designed the study, Wrote software for quantitative morphology analyses, Generated morphometric data sets, Performed analysis, Produced figures, Contributed to the manuscript; TB, Conceptualization, Software, Formal analysis, Writing—original draft, Writing—review and editing, Conceived of and designed the study, Wrote software for processing of confocal image stacks and quantitative morphology analyses, Contributed to the manuscript; CG, Formal analysis, Investigation, Methodology, Writing—review and editing, Executed wiring cost simulations, Contributed to the manuscript; EM, Conceptualization, Resources, Supervision, Funding acquisition, Writing—original draft, Writing—review and editing, Conceived of and designed the study, Wrote the original manuscript

### Author ORCIDs

Adriane G Otopalik, http://orcid.org/0000-0002-3224-6502

Eve Marder, http://orcid.org/0000-0001-9632-5448

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
