## [Decision Letter]

Thank you for submitting your article "Sloppy morphological tuning in identified neurons of the crustacean stomatogastric ganglion" for consideration by *eLife*. Your article has been favorably evaluated by David Van Essen (Senior Editor) and three reviewers, one of whom, Ronald L Calabrese (Reviewer #1), is a member of our Board of Reviewing Editors. The following individuals involved in review of your submission have agreed to reveal their identity: Hermann Cuntz (Reviewer #2); Davi D Bock (Reviewer #3).

The reviewers have discussed the reviews with one another and the Reviewing Editor has drafted this decision to help you prepare a revised submission.

Summary:

This very intriguing study provides an alternative scenario to the prevailing idea that neuronal arborization will conform to simple developmental rules that balance a minimization of total neurite length and soma-to-tip path lengths. Though the stomatogastric ganglion (STG) of crabs is not analogous to brain structures that have given rise to the prevailing concepts it does provide a stark contrast to this view. The spatial relationships between subtrees within individual neurons suggests that STG neurons grow to fill and tile distinct spatial fields in the neuropil. The authors then argue that because in the STG synaptic transmission is predominantly graded and slow, "in development, space-filling of the neuropil may outweigh the wiring economy that might be expected if decrement of synaptic events were critical to circuit function." They also argue that tiling the entire neuropil is advantageous for reception of paracrine modulatory input. The data presented is very high resolution whole mount dye fills of 4 types of STG neurons, two unique types that are singularly represented and two types that are represented by more than one copy. The analysis is highly quantitative and thoroughly analyzed statistically with appropriate techniques. The figures are not only clear but esthetically pleasing.

Essential revisions:

There are some concerns, especially with the minimal spanning trees analyses, that must be addressed, but overall this appears to be a very strong study with very high quality data.

1) The MST analyses should be redone as prescribed by reviewer 2. Once these analyses are redone the conclusions of the paper should be appropriated altered if needed.

2) Expansion of the range of considered *bf*'s should be performed as prescribed by reviewer 3.

3) An analysis of density distributions similar to the one by Jefferis et al. (Cell. 2007 Mar 23; 128(6): 1187-1203) should be performed on the Sholl analyses a prescribed by reviewer 3.

4) There is general concern that the Discussion does not consider all aspects pertinent to interpretation of the data analyses.

a) The neurons analyzed have both direct synaptic inputs and outputs throughout their neuritic arbors and there is limited data to definitively separate or lump these within the neuritic field. It is safest to assume that in STG neurons inputs and outputs are cheek by jowl given the work of King (1976 a, b) and Kilman and Marder (1996). There is no mention of this ambiguity or discussion concerning how inputs and outputs overlap or are segregated; the relevant papers are cited without mention of this aspect of their findings. Instead the Introduction and Discussion focus on modulatory inputs when there is even less data as to where the relevant receptors occur. It would be an interesting study to prove that input fields and output fields were separate or overlapping and relate that to the neuritic fields described, but here the issues must be aired, at least. Could this likely overlap of inputs and outputs be as important as graded transmission in determining STG neuronal morphology?

b) The STG is not the first nor the only previously observed exception to optimality of wiring. A more nuanced discussion here would help the manuscript as described by reviewer 3.

Reviewer #2:

The manuscript describes in exquisite detail the morphology of four cells in the stomatogastric ganglion (STG) of the crab – a type of study that as the authors point out hasn't been performed for many cell types to date with this amount of precision. The authors use a minimum spanning tree algorithm that I developed to check whether these neurons optimize wiring and they come to the conclusion that the neurons don't so much observe minimal amounts of cable and short conduction times to collect their inputs but rather fill the available neuropil. I was excited to find a system in which the considerations that we have found to be true for so many other systems (fly tangential cells, cortical pyramidal cells of many different types, cerebellar Purkinje cells, many different hippocampal cells, starburst amacrine cells in the retina, periglomerular neurons in the olfactory bulb etc., in fact, even all of NeuroMorpho.Org was subjected to the analysis in Cuntz et al. 2012 PNAS) would not hold.

Unfortunately the authors make a fundamental mistake in their analysis that overturns the results and conclusions. I will therefore focus my review on that part (Figure 10 and Figure 11) since the fact that crab STG neurons are not subjected strongly to optimal wiring constraints seems to be the main conclusion in the paper.

With our method of modelling the morphologies using minimum spanning trees (MSTs) it is important to realise that the dendrites collect their inputs from a set of given carrier locations (corresponding probably to a developmentally predestined subset of all synapses) and that connecting these optimally in terms of wiring describes a large proportion of the dendritic tree. Unfortunately, in most cases the locations of those carrier synapses are unknown (except for example for data from Adi Mizrahi on periglomerular neurons from the olfactory bulb that we used in Cuntz et al., 2012 PNAS). Therefore, one has to make some assumptions about which target locations should be connected by the MST. The authors here decided to take the termination points of the dendrites that they reconstructed as target points for the minimum spanning tree. This choice is not discussed. Connecting N termination points with an MST (constrained to be binary) will lead to termination points, branch points and continuation points (with one daughter). The resulting trees therefore will automatically have less termination points than the original trees. In binary trees (which Figure 4 shows the STG neurons mostly are), the number of branch points is equal to the number of termination points minus 1. In MSTs on N randomly distributed points with a *bf* = 0 we have shown previously that the distributed points divide into N/4 termination points, N/4 branch points and N/2 continuation points approximately (Cuntz et al., 2012 PNAS). This explains directly the findings in the present manuscript: Number of branches, total length, and average branch order are all dramatically underestimated here by the fact that only 25% of the correct number of target points were distributed (assuming *bf*=0, but otherwise certainly never more than about 40%).

In our previous work (Cuntz et al., 2008 PLoS CB; Cuntz et al., 2010 PLoS CB and others) we have set the number of target points in for example fly tangential cells such that the resulting MST had the correct number of branch points since we did not know the number or identity of target points from biology. In the case of the periglomerular neurons in (Cuntz et al., 2012 PNAS) we found that the number of branch points was indeed in numbers about 25% of the number of synapses marked by PSD95-GFP.

I am sorry for the lengthy explanation but I hope that it now becomes clear that the MSTs applied in this present manuscript need to be adjusted to match the complexity of the original trees. It is no surprise that the number of branch points and the total length are much lower in the MSTs if only the termination points (the tips) of the dendrites were used. After correcting this, the results will look very differently and maybe will show that the crab STG neurons also optimise wiring as so many others do (However analysis in Figure 9 indeed show a much larger tortuosity than in most other dendrites that I have looked at). If the results are still at odds with this claim after correction, then the conclusion from this paper holds.

Having said all this, I would like to point out that the study is very thoroughly done otherwise and after the correction of this issue could well be subjected to review one more time.

Reviewer #3:

This manuscript presents a high quality data set and accompanying analysis, and could readily be published with some revisions. However, I think the authors have in some ways left the job half-done. They show very nicely that crab stomatogastric ganglion neurites do not obey a standard wiring length optimality rule, but do not offer a positive alternative. In the last few sentences of the Discussion, they speculate that STG neurons maximize surface area in order to sample the "cocktail of peptides and amines" being released into this ganglion. This seems like a reasonably straightforward hypothesis to test using the existing data and analytical toolset the authors have marshalled. Specifically, does a geometrical model maximizing surface area allow for a better fit to the data than the Cuntz et al. spanning tree-based model? This type of analysis could be done in a follow-up paper, but it also seems within the authors' grasp here. (And, if a model maximizing surface area fits the observed arbors very closely, can the observed arbors really be considered 'sloppy'?)

A few additional high-level thoughts:

1) Would *bf* values well outside the previously used values of 0.1 and 0.85 give better fit to the STG data? The authors find that *bf* of 0.9 gives best fit to their data (subsection “Wiring Cost”, last paragraph); this is at one end of the parameter range explored. The authors show STG neurons are 'different'; they should explain why exploring a previously used parameter range is therefore acceptable.

2) The Sholl analysis of linearized arbors (Figure 2) is useful, but the linearization causes quantification of the density distribution of neurites to be lost. The neurons of each subtype could have variably branching patterns (as the authors show), while still having stereotyped density distributions in the STG. An analysis of density distributions similar to the one by Jefferis et al. (Cell. 2007 Mar 23; 128(6): 1187-1203) could be used to examine this.

3) In the Discussion, the authors write "This sloppy morphological tuning is in stark contrast to that which has been described in other neuron and circuit types." The wiring length optimization literature is at its most convincing when handling very large numbers of neurons (order 10^5 or more) traveling long distances (mm to cm or more). Van Essen's argument about cortical folding (Nature. 1997 Jan 23;385(6614):313-8) is an example at the extreme of this idea. At smaller length scales, for smaller numbers of neurons, wiring frequently looks highly suboptimal. For example, in the cat, retinogeniculate axons frequently make many branches which each independently target the same volume of LGN (Sur et al., J Neurophysiol. 1987 Jul;58(1):1-32), and in the mouse, motor neurons take extremely tortuous paths to cover the end plate (Lu et al., PLoS Biol. 2009 Feb 10;7(2):e32. doi: 10.1371/journal.pbio.1000032). The STG is not the first nor the only previously observed exception to optimality of wiring. A more nuanced discussion here would help the manuscript.

---

## [Author Response]

Essential revisions:

There are some concerns, especially with the minimal spanning trees analyses, that must be addressed, but overall this appears to be a very strong study with very high quality data.

1) The MST analyses should be redone as prescribed by reviewer 2. Once these analyses are redone the conclusions of the paper should be appropriated altered if needed.

We were grateful to have had the opportunity to discuss this issue directly with reviewer #2, who invented the MST analyses. From this reviewer’s comments, and our direct conversation, it was evident that our MST simulations should be tuned such that the resulting synthetic trees present a level of complexity similar to the actual STG neurons (this was not the case in the previously submitted manuscript). To address this issue, we generated carrier points from uniformly distributed points that spanned the elliptical volume of the actual STG neuron. We tuned the number of carrier points such that the branch point numbers of the resulting MSTs were within range of the actual STG neurons’ branch point numbers (all within 20%, but often within 5%). This revised simulation yielded MSTs with structural features (total wiring, branch order distributions, neurite length distributions, and tortuosity distributions) with realistic ranges, consistent with those observed in STG neurons. However, as is clear in Figure 11 and the supplements to Figure 11, no one *bf* comprehensively recapitulated these structural features of the STG neurons. Thus, our conclusions are left unchanged and we still interpret this to mean that STG neuronal morphology may not be constrained by the wiring optimization principles probed with the MST analysis.

2) Expansion of the range of considered bf's should be performed as prescribed by reviewer 3.

After discussion of the MST analyses with reviewer #2 and revision of the carrier points, the *bf* value range was changed to 0-0.6, at increments of 0.1. This range is sufficient, given this revised simulation, as the MSTs generated with this *bf* range present neurite length, branch order, and tortuosity distributions that range beyond the measured values of the actual neurons (this is clear in Figure 11 and the supplements to Figure 11). This addresses reviewer 3’s concern that the actual neurons appeared to fit best with *bf* values at one extreme of the previously chosen *bf* value range (this is no longer the case in the revised simulation).

3) An analysis of density distributions similar to the one by Jefferis et al. (Cell. 2007 Mar 23; 128(6): 1187-1203) should be performed on the Sholl analyses a prescribed by reviewer 3.

These new analyses are shown in Figure 3 and addressed in the last paragraph of the subsection “Expansive and Complex Neurite Trees”.

4) There is general concern that the Discussion does not consider all aspects pertinent to interpretation of the data analyses.

*a) The neurons analyzed have both direct synaptic inputs and outputs throughout their neuritic arbors and there is limited data to definitively separate or lump these within the neuritic field. It is safest to assume that in STG neurons inputs and outputs are cheek by jowl given the work of King (1976 a, b) and Kilman and Marder (1996). There is no mention of this ambiguity or discussion concerning how inputs and outputs overlap or are segregated; the relevant papers are cited without mention of this aspect of their findings. Instead the Introduction and Discussion focus on modulatory inputs when there is even less data as to where the relevant receptors occur. It would be an interesting study to prove that input fields and output fields were separate or overlapping and relate that to the neuritic fields described, but here the issues must be aired, at least. Could this likely overlap of inputs and outputs be as important as graded transmission in determining STG neuronal morphology?*

We now address this issue in the Discussion. In brief, we write that, “early electron microscopy studies describe the sparse distribution and tight apposition of synaptic input and outputs across the finer processes of the neurite tree, such that nearly every secondary process has both pre- and post-synaptic regions (King, 1976a, b). […] Regardless, the electrotonic compact nature of these neurons, along with this tight pairing of pre- and post-synaptic sites across the neurite tree, may effectively compensate for the complex and variable morphologies observed across animals.”

b) The STG is not the first nor the only previously observed exception to optimality of wiring. A more nuanced discussion here would help the manuscript as described by reviewer 3.

Please refer to our response to reviewer 3’s comments (below).

Reviewer #2:

The manuscript describes in exquisite detail the morphology of four cells in the stomatogastric ganglion (STG) of the crab – a type of study that as the authors point out hasn't been performed for many cell types to date with this amount of precision. The authors use a minimum spanning tree algorithm that I developed to check whether these neurons optimize wiring and they come to the conclusion that the neurons don't so much observe minimal amounts of cable and short conduction times to collect their inputs but rather fill the available neuropil. I was excited to find a system in which the considerations that we have found to be true for so many other systems (fly tangential cells, cortical pyramidal cells of many different types, cerebellar Purkinje cells, many different hippocampal cells, starburst amacrine cells in the retina, periglomerular neurons in the olfactory bulb etc., in fact, even all of NeuroMorpho.Org was subjected to the analysis in Cuntz et al. 2012 PNAS) would not hold.

Unfortunately the authors make a fundamental mistake in their analysis that overturns the results and conclusions. I will therefore focus my review on that part (Figure 10 and Figure 11) since the fact that crab STG neurons are not subjected strongly to optimal wiring constraints seems to be the main conclusion in the paper.

With our method of modelling the morphologies using minimum spanning trees (MSTs) it is important to realise that the dendrites collect their inputs from a set of given carrier locations (corresponding probably to a developmentally predestined subset of all synapses) and that connecting these optimally in terms of wiring describes a large proportion of the dendritic tree. Unfortunately, in most cases the locations of those carrier synapses are unknown (except for example for data from Adi Mizrahi on periglomerular neurons from the olfactory bulb that we used in Cuntz et al., 2012 PNAS). Therefore, one has to make some assumptions about which target locations should be connected by the MST. The authors here decided to take the termination points of the dendrites that they reconstructed as target points for the minimum spanning tree. This choice is not discussed. Connecting N termination points with an MST (constrained to be binary) will lead to termination points, branch points and continuation points (with one daughter). The resulting trees therefore will automatically have less termination points than the original trees. In binary trees (which Figure 4 shows the STG neurons mostly are), the number of branch points is equal to the number of termination points minus 1. In MSTs on N randomly distributed points with a bf = 0 we have shown previously that the distributed points divide into N/4 termination points, N/4 branch points and N/2 continuation points approximately (Cuntz et al., 2012 PNAS). This explains directly the findings in the present manuscript: Number of branches, total length, and average branch order are all dramatically underestimated here by the fact that only 25% of the correct number of target points were distributed (assuming bf=0, but otherwise certainly never more than about 40%).

In our previous work (Cuntz et al., 2008 PLoS CB; Cuntz et al., 2010 PLoS CB and others) we have set the number of target points in for example fly tangential cells such that the resulting MST had the correct number of branch points since we did not know the number or identity of target points from biology. In the case of the periglomerular neurons in (Cuntz et al., 2012 PNAS) we found that the number of branch points was indeed in numbers about 25% of the number of synapses marked by PSD95-GFP.

I am sorry for the lengthy explanation but I hope that it now becomes clear that the MSTs applied in this present manuscript need to be adjusted to match the complexity of the original trees. It is no surprise that the number of branch points and the total length are much lower in the MSTs if only the termination points (the tips) of the dendrites were used. After correcting this, the results will look very differently and maybe will show that the crab STG neurons also optimise wiring as so many others do (However analysis in Figure 9 indeed show a much larger tortuosity than in most other dendrites that I have looked at). If the results are still at odds with this claim after correction, then the conclusion from this paper holds.

Having said all this, I would like to point out that the study is very thoroughly done otherwise and after the correction of this issue could well be subjected to review one more time.

We are grateful to have received this feedback from the inventor of these analyses. After direct discussion with reviewer #2, we were able to resolve this issue by running a revised simulation with new carrier points (as described in Essential revisions point #1). In doing this, we were able to generate MSTs that reflected the complexity of the original trees (as described in reviewer #2’s comments above). We approached the revised simulation with an open mind toward its results and interpretation. However, the results of the revised simulation were such that our interpretation remains relatively unchanged. We did re-write parts of the Results and Discussion to better reflect the meaning of these analyses. It remains true that these analyses do, in fact, recapitulate the properties of numerous other neuron types. The fact that this is not the case for STG neurons does not undermine the wiring optimization theory, but simply demonstrates that wiring cost may not be a major factor in determining neuronal morphology in the STG.

Reviewer #3:

This manuscript presents a high quality data set and accompanying analysis, and could readily be published with some revisions. However, I think the authors have in some ways left the job half-done. They show very nicely that crab stomatogastric ganglion neurites do not obey a standard wiring length optimality rule, but do not offer a positive alternative. In the last few sentences of the Discussion, they speculate that STG neurons maximize surface area in order to sample the "cocktail of peptides and amines" being released into this ganglion. This seems like a reasonably straightforward hypothesis to test using the existing data and analytical toolset the authors have marshalled. Specifically, does a geometrical model maximizing surface area allow for a better fit to the data than the Cuntz et al. spanning tree-based model? This type of analysis could be done in a follow-up paper, but it also seems within the authors' grasp here. (And, if a model maximizing surface area fits the observed arbors very closely, can the observed arbors really be considered 'sloppy'?)

This last point is a good one. Although we have not chosen to address this follow-up question in this manuscript, it is something that we will consider moving forward.

A few additional high-level thoughts:

1) Would bf values well outside the previously used values of 0.1 and 0.85 give better fit to the STG data? The authors find that bf of 0.9 gives best fit to their data (subsection “Wiring Cost”, last paragraph); this is at one end of the parameter range explored. The authors show STG neurons are 'different'; they should explain why exploring a previously used parameter range is therefore acceptable.

This is addressed above, in response to Essential revision #2.

2) The Sholl analysis of linearized arbors (Figure 2) is useful, but the linearization causes quantification of the density distribution of neurites to be lost. The neurons of each subtype could have variably branching patterns (as the authors show), while still having stereotyped density distributions in the STG. An analysis of density distributions similar to the one by Jefferis et al. (Cell. 2007 Mar 23; 128(6): 1187-1203) could be used to examine this.

This type of analysis is now shown in Figure 3.

3) In the Discussion, the authors write "This sloppy morphological tuning is in stark contrast to that which has been described in other neuron and circuit types." The wiring length optimization literature is at its most convincing when handling very large numbers of neurons (order 10^5 or more) traveling long distances (mm to cm or more). Van Essen's argument about cortical folding (Nature. 1997 Jan 23;385(6614):313-8) is an example at the extreme of this idea. At smaller length scales, for smaller numbers of neurons, wiring frequently looks highly suboptimal. For example, in the cat, retinogeniculate axons frequently make many branches which each independently target the same volume of LGN (Sur et al., J Neurophysiol. 1987 Jul;58(1):1-32), and in the mouse, motor neurons take extremely tortuous paths to cover the end plate (Lu et al., PLoS Biol. 2009 Feb 10;7(2):e32. doi: 10.1371/journal.pbio.1000032). The STG is not the first nor the only previously observed exception to optimality of wiring. A more nuanced discussion here would help the manuscript.

We appreciate that reviewer #3 brought this area in the literature to our attention. We have addressed other exceptions to optimality of wiring in the Discussion, subsection “Sloppy Geometries that are Suitable for Circuit Function”. The studies we bring to light are perhaps the most intuitively similar to our case in the STG. In the third paragraph of the aforementioned subsection we mention the Lu et al., (2009) study, writing: “While this sloppy morphological tuning of STG neurons is notable, it is not the first exception to wiring optimality shown in the literature. […] As suggested by Lu et al. (2009), such case studies may not imply that wiring optimization principles are not at play, but may simply suggest that other factors constrain morphology in these nervous systems.”

In the sixth paragraph of the aforementioned subsection, we write of another more recent case that describes variability in dendritic tree morphology in one class of neocortical pyramidal neurons: “Yet, a recent study demonstrated a great deal of variability in dendritic morphology within one class of mammalian neocortical pyramidal neurons (Hamada et al., 2016). Interestingly, the distance between the soma and spike initiation zone co-varies with and compensates for variability in the dendritic tree. In this case, the pyramidal neuron achieves its target physiological properties by tuning ion channel distributions to the neuron’s geometry.”

By mentioning these works, we feel that we have provided a more thorough comparison of our findings to those in the current literature, and perhaps made it more clear that STG neurons are not the only exception to wiring optimality.